# An integrative analysis of the age-associated multi-omic landscape across cancers

Kasit Chatsirisupachai[1], Tom Lesluyes [2], Luminita Paraoan [3], Peter Van Loo [2] & João Pedro de Magalhães [1]✉

Age is the most important risk factor for cancer, as cancer incidence and mortality increase with age. However, how molecular alterations in tumours differ among patients of different age remains largely unexplored. Here, using data from The Cancer Genome Atlas, we comprehensively characterise genomic, transcriptomic and epigenetic alterations in relation to patients' age across cancer types. We show that tumours from older patients present an overall increase in genomic instability, somatic copy-number alterations (SCNAs) and somatic mutations. Age-associated SCNAs and mutations are identified in several cancer-driver genes across different cancer types. The largest age-related genomic differences are found in gliomas and endometrial cancer. We identify age-related global transcriptomic changes and demonstrate that these genes are in part regulated by age-associated DNA methylation changes. This study provides a comprehensive, multi-omics view of age-associated alterations in cancer and underscores age as an important factor to consider in cancer research and clinical practice.

[1] Integrative Genomics of Ageing Group, Institute of Life Course and Medical Sciences, University of Liverpool, Liverpool, UK. [2] The Francis Crick Institute, London, UK. [3] Department of Eye and Vision Science, Institute of Life Course and Medical Sciences, University of Liverpool, Liverpool, UK. ✉email: jp@senescence.info

Age is the biggest risk factor for cancer, as cancer incidence and mortality rates increase exponentially with age in most cancer types[1]. However, the relationship between ageing and molecular determinants of cancer remains to be characterised. Cancer arises through the interplay between somatic mutations and selection, in a Darwinian-like process[2,3]. Thus, apart from the accumulation of mutations with age[4–6], microenvironment changes during ageing could also play a role in carcinogenesis[2,7,8]. We therefore hypothesise that, due to the differences in selective pressures from tissue environmental changes with age, tumours arising from patients across different ages might harbour different molecular landscapes. Consequently, some molecular changes might be more or less common in older or younger patients.

Recently, several studies have investigated molecular differences in the cancer genome in relation to clinical factors, including gender[9,10] and race[11,12]. These studies demonstrated gender- and race-specific biomarkers, actionable target genes and provided clues to understanding the biology behind the disparities in cancer incidence, aggressiveness and treatment outcome across patients from different backgrounds. Although differences in genomic alterations between childhood and adult cancers have been systematically characterised[13,14], the age-related genomic landscape across adult cancers remains elusive. Specific age-associated molecular landscapes have been reported in the cancer genome of several cancer types, for example, glioblastoma[15], prostate cancer[16] and breast cancer[17]. However, these studies focused mainly on a single cancer type and only on some molecular data types.

Here, using data from The Cancer Genome Atlas (TCGA), we systematically investigate age-related differences in genomic instability (GI), somatic copy-number alterations (SCNAs), somatic mutations, pathway alterations, gene expression, and DNA methylation across various cancer types. We show that, in general, GI and mutations frequency increase with age. We identify several age-associated genomic alterations in cancers, particularly in low-grade glioma and endometrial carcinoma. Moreover, we also demonstrate that age-related gene expression changes are partly controlled by age-related DNA methylation changes and that these changes are linked to numerous biological processes.

## Results

### Association between age and genomic instability, loss of heterozygosity, and whole-genome duplication.
To gain insight into the role of patient age into the somatic genetic profile of tumours, we evaluated associations between patient age and genomic features of tumours in TCGA data (Table 1, Supplementary Data 1). We first derived GI scores, calculated as the genome fraction (percent-based) that does not fit the ground state, defined as 2 for tumours that have not undergone whole-genome duplication (WGD), and 4 for tumours that have undergone WGD (Methods). Using multiple linear regression adjusting for gender, race, and cancer type, we found that GI scores increase with age in pan-cancer data (adj. R-squared = 0.35, $p$ value = $5.98 \times 10^{-7}$) (Fig. 1a). We next applied simple linear regression to investigate the relationship between GI scores and age for each cancer type. Cancer types with a significant association (adj. $p$ value < 0.05) were further adjusted for clinical variables. We found a significant positive association between age and GI score in seven cancer types (adj. $p$ value < 0.05) (Fig. 1b, Supplementary Fig. 1a and Supplementary Data 2). Cancer types with the strongest significant positive association were low-grade glioma, ovarian cancer, endometrial cancer and sarcoma. This result indicates that the level of GI increases with the age of cancer patients in several cancer types.

Genomic loss of heterozygosity (LOH) refers to the irreversible loss of one parental allele, causing an allelic imbalance, and priming the cell for another defect at the other remaining allele of the respective genes[18]. To investigate whether there is an association between patients' age and LOH, we quantified percent genomic LOH. By using simple linear regression, we found a significant positive association between age and pan-cancer percent genomic LOH ($p$ value = $1.20 \times 10^{-21}$). However, this association was no longer significant in a multiple linear regression analysis (adj. R-squared = 0.32, $p$ value = 0.289) (Fig. 1c). Thus, it is likely that this association might be cancer type-specific. We then performed a linear regression between age and percent genomic LOH for each cancer type. Six cancer types showed a positive association between age and percent genomic LOH (adj. $p$ value < 0.05) (Fig. 1d, Supplementary Fig. 1b, and Supplementary Data 3). The strongest positive associations were found in low-grade glioma and endometrial cancer (adj. $p$ value < 0.05), corroborating with the increase in GI score with age. On the other hand, lung adenocarcinoma, oesophageal and liver cancer demonstrated a negative correlation between percent genomic LOH and age (adj. $p$ value < 0.05). This negative correlation might be due to the difference in the distribution of age of samples with smoking status (lung adenocarcinoma and oesophageal cancer), race (oesophageal cancer) and tumour grade (liver cancer) (Supplementary Fig. 2), yet other unexplained factors might also contribute to the higher LOH level in younger patients in these three cancer types.

WGD is important in increasing the adaptive potential of the tumour and has been linked with a poor prognosis[19–21]. We investigated the relationship between age and WGD using logistic regression. In a pan-cancer analysis, we found a small increase in the probability that WGD occurs with age, using multiple logistic regression accounting for gender, race, and cancer type (odds ratio per year (OR) = 1.0066, 95% confidence interval (CI) = 1.0030–1.0103, $p$ value = $3.84 \times 10^{-4}$) (Fig. 1e). For the cancer-specific analysis, a significant positive association was found in ovarian and endometrial cancer (OR = 1.0320 and 1.0248, 95% CI = 1.0151–1.0496 and 1.0024–1.0483, adj. $p$ value = $4.68 \times 10^{-4}$ and 0.049, respectively) (Fig. 1e and Supplementary Data 4), indicating that tumours from older patients are more likely to have doubled their genome. Taken together, the findings indicate that tumours from patients with an increased age tend to harbour a more unstable genome and a higher level of LOH in several cancer types. Notably, the strongest association between age and an increase in genome instability, LOH, and WGD was evident in endometrial cancer, suggesting potential disparities in the cancer genome landscape with age in this cancer type.

### Age-associated somatic copy-number alterations.
We used GISTIC2.0 to identify recurrently altered focal- and arm-level SCNAs[22]. We calculated the SCNA score, as a representation of the level of SCNA occurring in a tumour[12,23]. For each tumour, the SCNA score was calculated at three different levels: focal-, arm- and chromosome-level, and the overall score calculated from the sum of all three levels. We used simple linear regression to identify the association between age and overall SCNA scores. Cancer types that displayed a significant association were further adjusted for clinical variables. Consistent with the GI score results described above, the strongest positive association between age and overall SCNA scores was found in low-grade glioma, ovarian and endometrial cancers. Other cancer types for which a positive association between age and overall SCNA score was observed were thyroid cancer and clear cell renal cell carcinoma (adj. $p$ value < 0.05). On the other hand, lung adenocarcinoma is the only cancer type exhibiting a negative association between overall

**Table 1 Summary of TCGA cancer type and number of samples used in each analysis.**

| Cancer type | Abbreviation | GI, LOH, WGD | SCNAs | Mutations (hypermutated and MSI-H removed) | Pathway alterations | Gene expression | DNA methylation |
|---|---|---|---|---|---|---|---|
| Adrenocortical carcinoma | ACC | 89 | 89 | 89 (88) | 76 | 77 | 78 |
| Bladder urothelial carcinoma | BLCA | 370 | 369 | 369 (364) | 361 | 366 | 370 |
| Breast invasive carcinoma | BRCA | 1015 | 1011 | 954 (946) | 922 | 1011 | 719 |
| Cervical squamous cell carcinoma and endocervical adenocarcinoma | CESC | 287 | 287 | 271 (263) | 264 | 284 | 287 |
| Cholangiocarcinoma | CHOL | 35 | 35 | 35 (35) | 35 | 35 | 35 |
| Colon adenocarcinoma | COAD | 411 | 411 | 374 (303) | 323 | 410 | 278 |
| Lymphoid neoplasm diffuse large B-cell lymphoma | DLBC | 42 | 42 | 32 (32) | 32 | 42 | 42 |
| Oesophageal carcinoma | ESCA | 176 | 176 | 176 (174) | 165 | 175 | 176 |
| Glioblastoma multiforme | GBM | 489 | 489 | 356 (354) | 116 | 137 | 259 |
| Head and neck squamous cell carcinoma | HNSC | 489 | 489 | 472 (469) | 459 | 481 | 489 |
| Kidney chromophobe | KICH | 66 | 66 | 66 (66) | 65 | 66 | 66 |
| Kidney renal clear cell carcinoma | KIRC | 496 | 496 | 343 (343) | 331 | 493 | 296 |
| Kidney renal papillary cell carcinoma | KIRP | 228 | 228 | 222 (222) | 215 | 228 | 213 |
| Acute myeloid leukaemia | LAML | 126 | 121 | 55 (54) | 101 | 102 | 121 |
| Brain lower grade glioma | LGG | 488 | 488 | 484 (484) | 482 | 488 | 488 |
| Liver hepatocellular carcinoma | LIHC | 355 | 355 | 342 (340) | 334 | 349 | 355 |
| Lung adenocarcinoma | LUAD | 460 | 460 | 456 (438) | 446 | 456 | 402 |
| Lung squamous cell carcinoma | LUSC | 460 | 460 | 444 (437) | 426 | 457 | 336 |
| Mesothelioma | MESO | 82 | 82 | 77 (77) | 77 | 82 | 82 |
| Ovarian serous cystadenocarcinoma | OV | 556 | 556 | 397 (395) | 173 | 288 | 545 |
| Pancreatic adenocarcinoma | PAAD | 133 | 133 | 130 (129) | 113 | 127 | 132 |
| Pheochromocytoma and Paraganglioma | PCPG | 165 | 157 | 165 (164) | 154 | 165 | 165 |
| Prostate adenocarcinoma | PRAD | 434 | 434 | 434 (432) | 425 | 434 | 434 |
| Rectum adenocarcinoma | READ | 152 | 152 | 132 (125) | 109 | 151 | 95 |
| Sarcoma | SARC | 229 | 229 | 213 (211) | 209 | 227 | 229 |
| Skin cutaneous melanoma | SKCM | 434 | 434 | 432 (340) | 332 | 433 | 434 |
| Stomach adenocarcinoma | STAD | 388 | 388 | 385 (345) | 340 | 365 | 341 |
| Testicular germ cell tumours | TGCT | 129 | 129 | 124 (124) | 123 | 129 | 129 |
| Thyroid carcinoma | THCA | 260 | 260 | 249 (248) | 244 | 259 | 258 |
| Thymoma | THYM | 76 | 76 | 76 (76) | 73 | 73 | 76 |
| Uterine corpus endometrial carcinoma | UCEC | 434 | 434 | 421 (282) | 406 | 432 | 360 |
| Uterine carcinosarcoma | UCS | 52 | 52 | 52 (51) | 52 | 52 | 52 |
| Uveal melanoma | UVM | 72 | 72 | 72 (72) | 72 | 72 | 72 |
| Total | | 9678 | 9660 | 8899 (8448) | 8055 | 8946 | 8414 |

SCNA score and age (Fig. 2a, Supplementary Fig. 3a, and Supplementary Data 5), possibly due to the presence of current smokers in younger lung adenocarcinoma patients (Supplementary Fig. 2a). When we analysed only non-smokers, there was no significant association between age and overall SCNA score (Supplementary Fig. 4a). However, the significant negative association was found when we analysed only current reformed smokers and only current smokers (Supplementary Fig. 4b, c), thus other unexplained factors apart from smoking status might also contribute to this higher SCNA score in younger smokers.

The different SCNA classes (focal- and chromosome/arm-level) may arise through different biological mechanisms[12,21]. Therefore, we separately analysed the association between age and focal- and chromosome/arm-level SCNA scores. Most cancer types that showed a significant relationship between age and overall SCNA score also had an association between age and both chromosome/arm-level and focal-level SCNA scores (Fig. 2b, c, Supplementary Fig. 3b, c, and Supplementary Data 5). The only exception was sarcoma, with a significant association between age and chromosome/arm-level but not with focal-level and overall SCNA scores.

We next identified the chromosomal arms that tend to be gained and lost more often with age, for 25 cancer types with sufficient samples (at least 100 tumours, Table 1). We conducted logistic regression on the significant recurrently gained and lost arms that were identified by GISTIC2.0 for each cancer type. The significant associations between age and chromosomal arm gains and losses are shown in Fig. 2d, e, respectively (adj. $p$ value < 0.05) (Supplementary Fig. 5, Supplementary Data 6). Gains of chromosome 7p, 7q, 20p, and 20q significantly increased with age in several cancer types including two types of gliomas, low-grade glioma and glioblastoma. On the other hand, the gain of

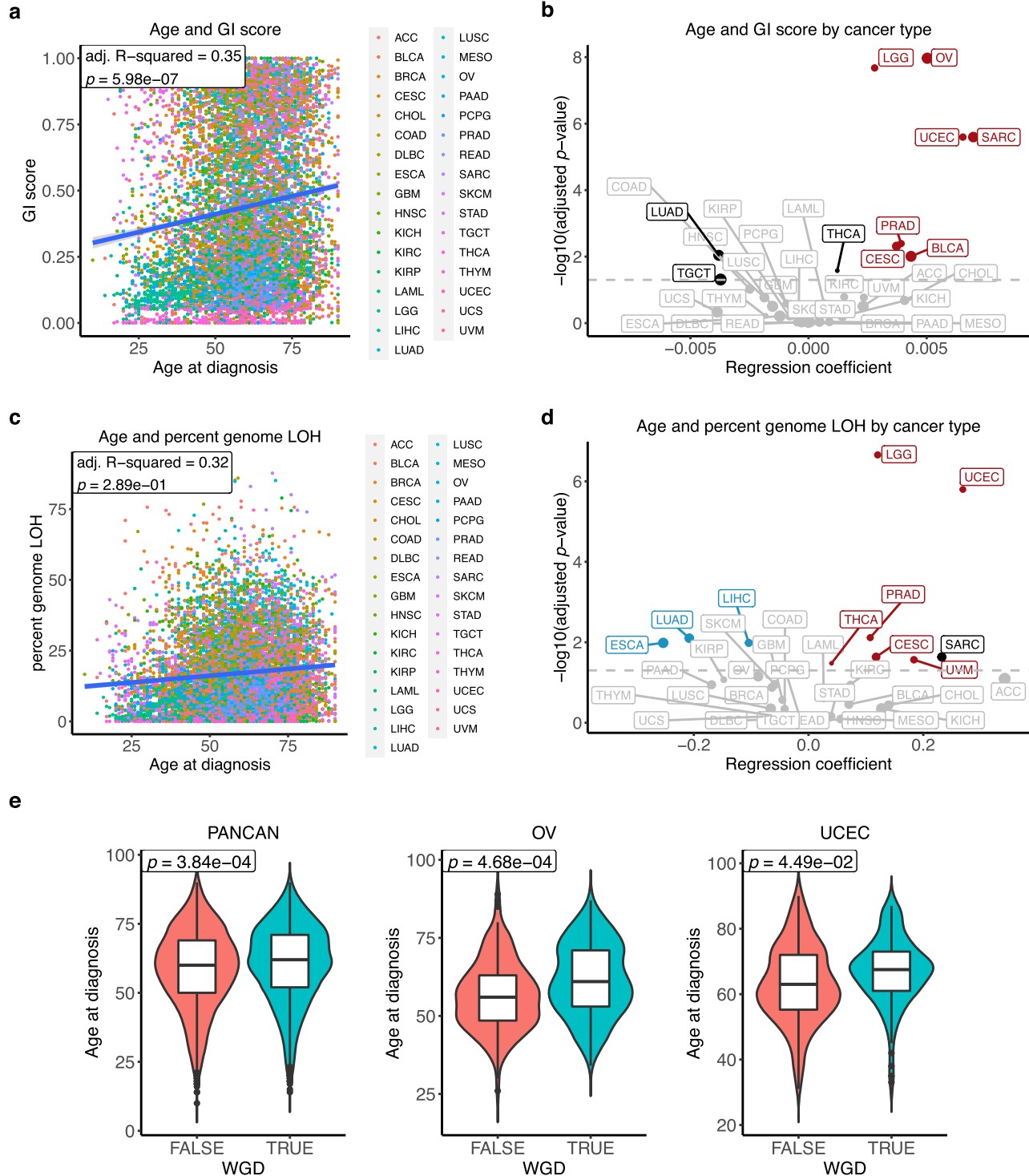

chromosome 10p decreased with increased age in gliomas (Fig. 2d, f). For the arm losses, there was an increased occurrence of loss in 11 arms with advanced age in endometrial cancer (Fig. 2e, g), consistent with a higher GI and LOH with age in this cancer type. These arms included 9p and 17p, containing tumour suppressor genes *CDKN2A* and *TP53*, respectively. Low-grade glioma and ovarian cancer, two other cancer types for which we found the highest significant association between age and SCNA scores, also exhibited a significant increase or decrease in losses with age in multiple arms (Fig. 2e, f, Supplementary Fig. 5). We also observed that losses of chromosome 10p and 10q increased

with age in gliomas. Recurrent losses of chromosome 10 (containing *PTEN*), together with gains of chromosome 7 (containing *EGFR*) are important features in IDH-wild-type (IDH-WT) gliomas[24]. This type of glioma was more common in older patients, whereas IDH-mutant gliomas were predominantly found in younger patients. Apart from gliomas and endometrial cancer, arm-level gains and losses in other cancer types are also related to known cancer-driver genes. For instance, we found an increased incidence in the loss of chromosome 13q (harbouring *RB1*) with age in thyroid cancer. Gains of chromosome 12 (containing the *KRAS* oncogene), increased with age in ovarian

**Fig. 1 Association between cancer patients' age and genomic instability (GI) score, percent genomic loss-of-heterozygosity (LOH) and whole-genome duplication events (WGD).** **a** Association between age and pan-cancer GI score. Dots are coloured by cancer type. Multiple linear regression R-squared and *p* value are shown in the figure. Multiple-hypothesis testing correction was not performed (single test). **b** Association between age and cancer type-specific GI score. Linear regression coefficients and significant values are shown in the figure. Multiple-hypothesis testing correction was done using Benjamini–Hochberg procedure. Cancers with a significant positive association between age and GI score after using multiple linear regression (adj. *p* value < 0.05) are highlighted in red. Cancers with a significant association in simple linear regression but not significant after using multiple linear regression are showed in black. The grey line indicates adj. *p* value = 0.05. Dot size is proportional to median GI score. **c** Association between age and pan-cancer percent genomic LOH. Dots are coloured by cancer type. Multiple linear regression R-squared and *p* value are shown in the figure. Multiple-hypothesis testing correction was not performed (single test). **d** Association between age and cancer type-specific percent genomic LOH. Linear regression coefficients and significant values are shown in the figure. Multiple-hypothesis testing correction was done using Benjamini–Hochberg procedure. Cancers with a significant positive and negative association between age and percent genomic LOH after using multiple linear regression are highlighted in red and blue, respectively. Cancer with a significant association in simple linear regression but not significant after using multiple linear regression is showed in black. The grey line indicates adj. *p* value = 0.05. Dot size is proportional to median percent genomic LOH. **e** Association between age and WGD events in pan-cancer (FALSE *n* = 5313, TRUE *n* = 4365 samples), OV (FALSE *n* = 207, TRUE *n* = 349 samples), and UCEC (FALSE *n* = 294, TRUE *n* = 140 samples). Multiple logistic regression *p* values were indicated in the figure. Multiple-hypothesis testing correction was done using Benjamini–Hochberg procedure. The middle bar of the boxplot is the median. The box represents interquartile range (IQR), 25th to 75th percentile. Whiskers represent a distance of 1.5 × IQR. TCGA cancer type acronyms and their associated name are provided in Table 1.

cancer. Indeed, while we can explain some of the age-associated chromosomal arm alterations, further closer inspection of arm-level alterations is required to fully explain why some specific arms are more or less frequently gained or lost as a function of age in particular cancer types.

We further examined age-associated recurrent focal SCNAs. Applying a similar logistic regression, we identified recurrent focal SCNAs associated with the age of the patients for each cancer type. In total, we found 113 significant age-associated regions, including 67 gains across 10 cancer types and 46 losses across 9 cancer types (adj. *p* value < 0.05) (Fig. 3a, Supplementary Data 7). In accordance with the arm-level result, the highest number of significant regions was found in endometrial cancer (23 gains and 25 losses), followed by ovarian cancer (13 gains and 2 losses) and low-grade glioma (9 gains and 5 losses) (Fig. 3b, c, Supplementary Fig. 6).

To further investigate the impact of these SCNAs, we studied the correlation between the SCNAs and gene expression for tumours that have both types of data using Pearson correlation. In total, 81 genes in the list of previously identified cancer-driver genes[25–27] (Supplementary Data 8) were presented in at least one significant age-associated focal region in at least one cancer type and showed a significant correlation between SCNA and gene expression (adj. *p* value < 0.05) (Fig. 3d). For example, regions showing an increased gain with age in endometrial cancer included 1q22, where the gene *RIT1* is located in (OR = 1.0355, 95% CI = 1.0151–1.0571, adj. *p* value = 0.0018) (Fig. 3c, e). The Ras-related GTPases *RIT1* has been reported to be highly amplified and correlated with poor survival in endometrial cancer[28]. Therefore, an increased incidence of *RIT1* gains with age might relate to a poor prognosis in older patients. The 16p13.3 loss increased in frequency in older endometrial cancer patients (OR = 1.0335, 95% CI = 1.0048–1.0640, adj. *p* value = 0.0328). This region contains the p53 coactivator gene *CREBBP*. The gain of 8q24.21 (harbouring the oncogene *MYC*) decreased with patient age in low-grade glioma (OR = 0.9737, 95% CI = 0.9541–0.9927, adj. *p* value = 0.0128) and ovarian cancer (OR = 0.9729, 95% CI = 0.9553–0.9904, adj. *p* value = 0.0063) (Fig. 3d, e). In addition, in low-grade glioma, we found an increased incidence of 9p21.3 loss with age (OR = 1.0332, 95% CI = 1.0174–1.0496, adj. *p* value = 0.00017). This region contains the cell cycle-regulator genes *CNKN2A* and *CDKN2B* (Fig. 3b, d, e). The full list of age-associated focal regions across cancer types and the correlation between SCNA status and gene expression can be found in Supplementary Data 7. Taken together, our analysis demonstrates the association between age and SCNAs

across cancer types. We also identified age-associated arm-level and focal regions, and these regions harboured several known cancer-driver genes. Our results suggest a possible contribution of different SCNA events in cancer initiation and progression of patients with different ages.

**Age-associated somatic mutations in cancer.** The increase in mutation burden with age is well-established[4–6]. This age-related mutation accumulation is in part explained by a clock-like mutational process, spontaneous deamination of 5-methylcytosine to thymine[5]. As expected, we confirmed the positive association between age and mutation load (somatic non-silent SNVs and indels) in the pan-cancer cohort using multiple linear regression adjusting for gender, race, and cancer type (adj. R-squared = 0.53, *p* value = $1.41 \times 10^{-37}$) (Fig. 4a). In cancer-specific analyses, 18 cancer types exhibited a significant relationship between age and mutation load using linear regression (adj. *p* value < 0.05) (Supplementary Fig. 7a, Supplementary Data 9). This increase in mutation load was mainly contributed by C>T mutations, as we found a positive association between the fraction of C>T mutations and age (regression coefficient = 0.058, *p* value = $8.57 \times 10^{-7}$) in a pan-cancer analysis. Conversely, the fraction of C>A mutations was negatively associated with age (regression coefficient = −0.065, *p* value = $8.84 \times 10^{-10}$) (Supplementary Data 10), concordant with a previous report[4]. We also examined, for each cancer type, the association between age and fraction contribution of each substitution class. Consistent with the pan-cancer analysis, C>T mutations showed a significant positive association with age in six cancer types, whereas C>A mutations had a significant negative association with age in three cancer types. (Supplementary Fig. 7b, Supplementary Data 10).

Only endometrial cancer showed a negative correlation between mutation burden and age (Supplementary Fig. 7a). We observed a high proportion of hypermutated tumours (>1000 non-silent mutations per exome) from younger endometrial cancer patients. Thirteen out of 38 tumours (34%) from the younger patients (age ≤ 50) were hypermutated tumours, while there were only 42 hypermutated tumours among the 383 tumours from older patients (11%) (two-sided Fisher's exact, *p* value = 0.0003) (Fig. 4b). Microsatellite instability (MSI) is a unique molecular alteration caused by defects in DNA mismatch repair[29,30]. The MSI-high (MSI-H) tumours occur as a subset of high mutation burden tumours[31]. We investigated whether high mutation loads in endometrial cancer from young patients were due to the presence of MSI-H tumours. Using multiple logistic regression, we

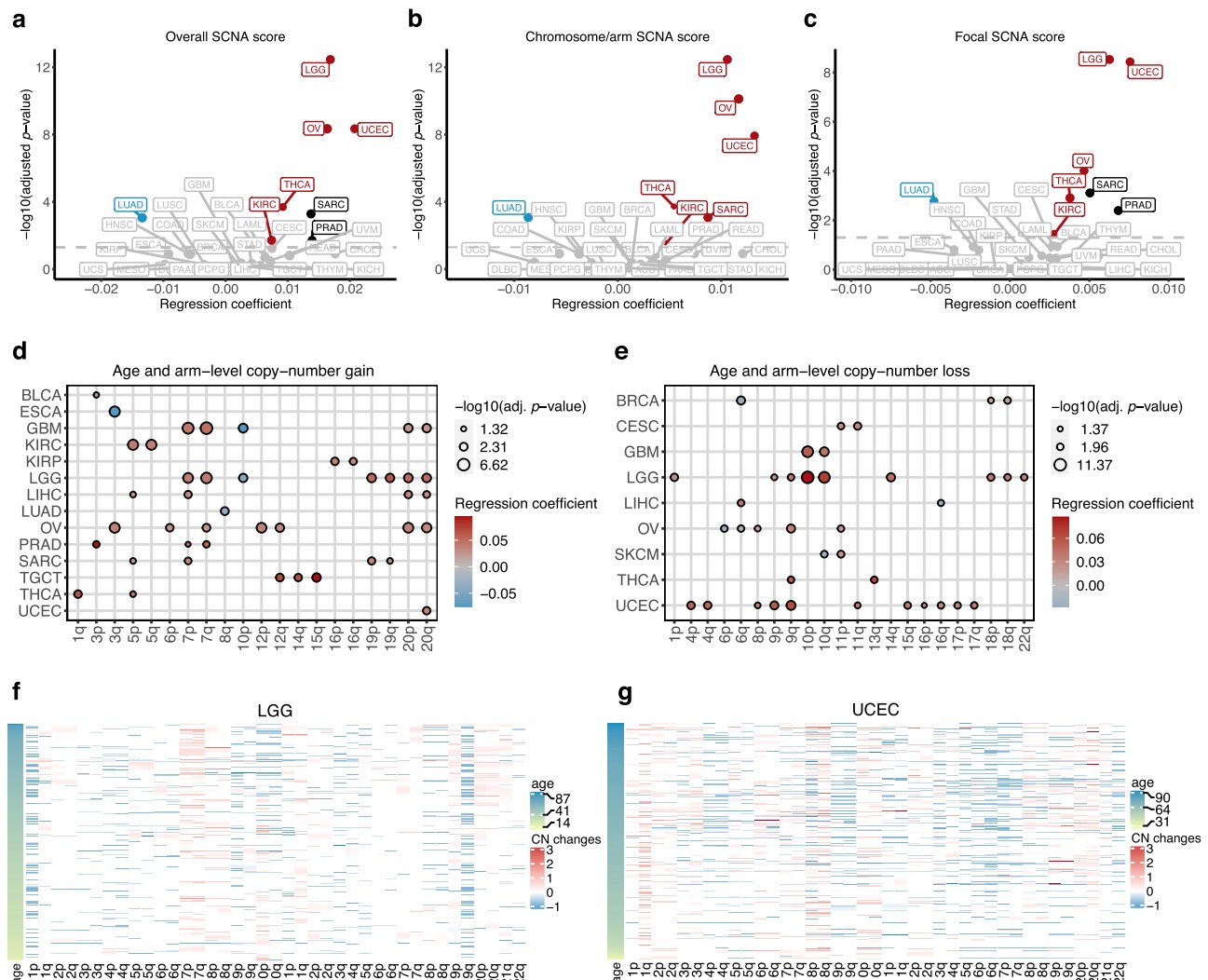

**Fig. 2 Association between cancer patients' age and somatic copy-number alterations (SCNAs).** Volcano plot representing the association between age and (**a**) overall, (**b**) focal-level and (**c**) chromosome/arm-level SCNA scores. Linear regression coefficients and significant values are shown. Multiple-hypothesis testing correction was done using Benjamini–Hochberg procedure. Cancers with a significant positive and negative association between age and SCNA score after using multiple linear regression (adj. *p* value < 0.05) are highlighted in red and blue, respectively. Cancers with a significant association in simple linear regression but not significant after using multiple linear regression are showed in black. The grey line indicates adj. *p* value = 0.05. Dot size is proportional to median SCNA score. **d**, **e** The left and right dot plots show the association between age and arm-level copy-number gains and copy-number losses. Multiple logistic regression coefficients and significant values are shown. Multiple-hypothesis testing correction was done using Benjamini–Hochberg procedure. Circle size corresponds to the significant level, red and blue represent positive and negative associations, respectively. **f**, **g** Heatmaps represent arm-level copy-number alterations in LGG and UCEC, respectively. Samples are sorted by age. Colours represent copy-number changes from GISTIC2.0, blue denotes loss and red corresponds to gain. TCGA cancer type acronyms and their associated name are provided in Table 1.

found that MSI-H tumours were associated with younger endometrial cancer (OR = 0.9751, 95% CI = 0.9531–0.9971, *p* value = 0.0264) (Fig. 4c). Another source of hypermutation in cancer is defective DNA polymerase proofreading due to mutations in polymerase ε (*POLE*) or polymerase δ (*POLD1*) genes[32,33]. We showed that mutations in *POLE* (OR = 0.9690, 95% CI = 0.9422–0.9959, *p* value = 0.0243) and *POLD1* (OR = 0.9573, 95% CI = 0.9223–0.9925, *p* value = 0.0177) were both more prevalent in younger endometrial cancer patients (Fig. 4d). Indeed, when we excluded tumours with MSI-H and tumours containing *POLE/POLD1* mutations from the analysis, we found a significant positive association between mutation burden and age in endometrial cancer (adj. R-squared = 0.12, *p* value = 0.00138) (Supplementary Fig. 7c). Therefore, the negative correlation between age and mutation loads in endometrial cancer could be

explained by the presence of hypermutated tumours in younger patients, which are associated with MSI-H and *POLE/POLD1* mutations. Previous studies on *POLE* and MSI-H subtypes in hypermutated endometrial tumours revealed that these subtypes associated with a better prognosis when compared with the copy-number high subtype[34–36]. Together with our SCNA results, younger endometrial cancer patients are likely to associate with a *POLE* and MSI-H subtypes, high mutation rate and better survival, whilst tumours from older patients are characterised by many SCNAs and are generally associated with a worse prognosis, indicating differences between age-related subtypes in endometrial cancer. We extended the age and MSI-H analysis to other cancer types known to have a high prevalence of MSI-H tumours, including colon, rectal, and stomach cancers[29]. Only in stomach cancer we found an association between older age and the presence

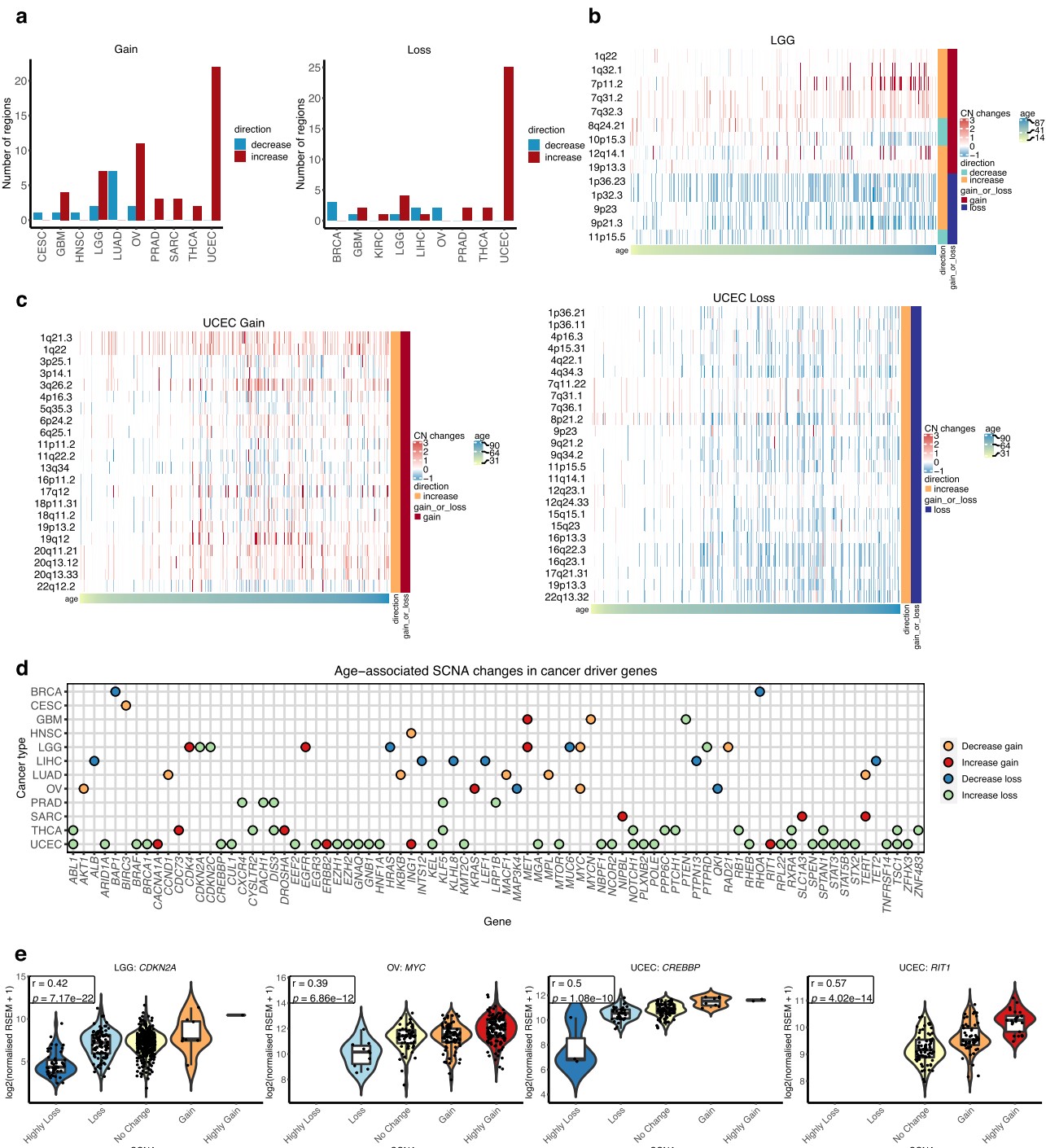

**Fig. 3 Association between cancer patients' age and focal-level SCNAs. a** Number of gained and deleted focal regions that showed a significant association with age per cancer type (multiple logistic regression, adj. p value < 0.05). Multiple-hypothesis testing correction was done using Benjamini–Hochberg procedure. Heatmap showing age-associated focal-level SCNAs in (**b**) LGG and (**c**) UCEC. Samples are sorted by age. Colours represent copy-number changes from GISTIC2.0, blue denotes loss and red corresponds to gain. The gain_or_loss legend demonstrates that the region is recurrently gained or deleted. The direction legend shows whether the gain/loss of the region increases or decreases with age. **d** Age-associated SCNA changes in cancer-driver genes. Cancer-driver genes located in the age-associated focal regions are plotted by cancer type. Colours of the dot represent the condition of the focal region where the gene located in as follows: blue—decrease loss; green—increase loss; yellow—decrease gain; and red—increase gain with age. **e** The effect of copy-number changes on gene expression of *CDKN2A* in LGG (Highly Loss n = 50, Loss n = 102, No Change n = 327, Gain n = 8, Highly Gain n = 1 samples), *MYC* in OV (Highly Loss n = X, Loss n = 10, No Change n = 56, Gain n = 82, Highly Gain n = 140 samples), *CREBBP* (Highly Loss n = 3, Loss n = 36, No Change n = 101, Gain n = 8, Highly Gain n = 2 samples) and *RIT1* (No Change n = 72, Gain n = 53, Highly Gain n = 25 samples) in UCEC. These are examples of genes with age-associated changes in SCNAs. Violin plots show the log2 (normalised expression + 1) of samples grouped by their SCNA status. Pearson correlation coefficient r and p value (two-sided test) are shown in the figures. The middle bar of the boxplot is the median. The box represents interquartile range (IQR), 25–75th percentile. Whiskers represent a distance of 1.5 × IQR. TCGA cancer type acronyms and their associated name are provided in Table 1.

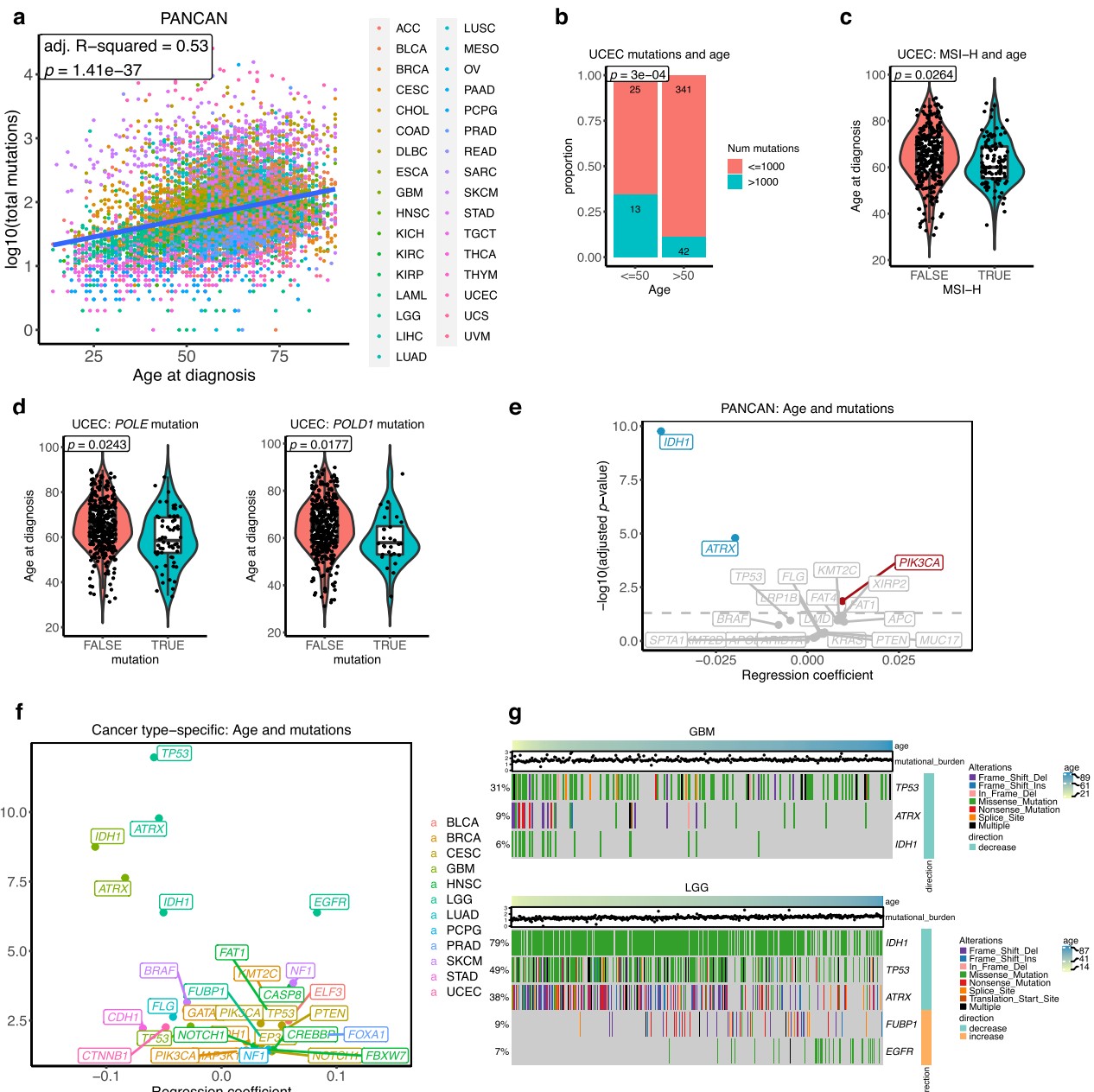

**Fig. 4 Association between cancer patients' age and somatic mutations. a** Association between patient's age and mutation burden in pan-cancer. Adjusted R-squared and p value from multiple linear regression analysis are presented. Multiple-hypothesis testing correction was not performed (single test). **b** The proportion of hypermutated tumours (>1000 mutations/exome) in young (age ≤ 50) and old (age > 50) UCEC. The statistical significant (p value) was calculated using two-sided Fisher's exact test. **c** The association between age and MSI-H in UCEC. FALSE n = 286, TRUE n = 106 samples. The statistical significance (p value) was calculated from the multiple logistic regression adjusting for clinical variables. **d** The association between age and POLE/POLD1 mutations in UCEC. POLE FALSE n = 359, TRUE n = 62 samples. POLD1 FALSE n = 392, TRUE n = 29 samples. The statistical significance (p value) was calculated from the multiple logistic regression adjusting for clinical variables. The middle bar of the boxplot is the median. The box represents interquartile range (IQR), 25–75th percentile. Whiskers represent a distance of 1.5 × IQR. **e** A pan-cancer association between age and mutations. Multiple logistic regression coefficient and significant values are shown. Multiple-hypothesis testing correction was done using Benjamini–Hochberg procedure. Genes with a significant positive and negative association between age and somatic mutations after using multiple logistic regression (adj. p value < 0.05) are highlighted in red and blue, respectively. **f** Summary of the cancer type-specific association between age and mutations. Multiple logistic regression coefficient and significant values are shown. Multiple-hypothesis testing correction was done using Benjamini–Hochberg procedure. Only genes with a significant association (adj. p value < 0.05) are shown in the figure. A colour code is provided to denote the cancer type where the association between age and gene mutation was found. **g** Heatmap showing age-associated mutations in GBM and LGG. Samples are sorted by age. Colours represent types of mutation. The right annotation legend indicates the direction of change, increase or decrease mutations with age. The mutation burden of samples is presented in the dot above the heatmap. TCGA cancer type acronyms and their associated name are provided in Table 1.

of MSI-H tumours, as reported previously[37] (OR = 1.0392, 95% CI = 1.0091–1.0720, $p$ value = 0.01, Supplementary Fig. 8a). When we further examined associations between age and mutations in *POLE* and *POLD1* in other cancers apart from endometrial cancer, no significant associations were observed (Supplementary Fig. 8b).

Although the increase in mutation load with age in cancer is well studied[4,31], differences in mutation rate in particular genes as a function of age across cancer types are largely unknown. To better understand this, we conducted logistic regression to investigate genes that are more or less likely to be mutated with an increased age. To prevent potential biases caused by hypermutated tumours, we restricted the analysis to samples with <1000 non-silent mutations per exome and those that are not MSI-H tumours (Table 1). We first investigated associations between age and pan-cancer gene-level mutations. Using multiple logistic regression correcting for gender, race, and cancer type, mutations in *IDH1* (OR = 0.9608, 95% CI = 0.9497–0.9719, adj. $p$ value = 1.73 × 10$^{-10}$) and *ATRX* (OR = 0.9804, 95% CI = 0.9725–0.9884, adj. $p$ value = 1.60 × 10$^{-5}$) showed a negative association with age. On the other hand, mutations in *PIK3CA* were more common in older individuals (OR = 1.0096, 95% CI = 1.0035–1.0158, adj. $p$ value = 0.0139) (Fig. 4e). We next identified genes exhibiting mutation rate differences associated with age in a cancer-specific manner, in 24 cancer types with at least 100 samples (Table 1). Using logistic regression, we identified 31 mutations in 12 cancer types that increased or decreased as a function of the patients' age (adj. $p$ value < 0.05) (Fig. 4f, g, Supplementary Fig. 9 and Supplementary Data 11). The most striking negative associations between mutations and age in low-grade glioma and glioblastoma were found in *IDH1* (OR = 0.9509 and 0.8962, 95% CI = 0.9328–0.9686 and 0.8598–0.9291, adj. $p$ value = 4.12 × 10$^{-7}$ and 1.78 × 10$^{-9}$, respectively), *ATRX* (OR = 0.9471 and 0.9120, 95% CI = 0.9310–0.9628 and 0.8913–0.9466, adj. $p$ value = 1.67 × 10$^{-10}$ and 2.33 × 10$^{-8}$, respectively), and *TP53* (OR = 0.9431 and 0.9736, 95% CI = 0.9274–0.9582 and 0.9564–0.9905, adj. $p$ value = 1.08 × 10$^{-12}$ and 5.16 × 10$^{-3}$, respectively). Our observation was consistent with the fact that the median age of IDH-mutants is younger than IDH-WT gliomas. Patients carrying *IDH1* mutations generally had a longer survival than IDH-WT patients[38]. Previous studies also reported that *IDH1* mutations often co-occurred with *ATRX* and *TP53* mutations, and mutations in these three genes were more prevalent in gliomas without *EGFR* mutations[15,39]. Indeed, we found that *EGFR* mutations were more common in older low-grade glioma patients (OR = 1.0865, 95% CI = 1.0525–1.1258, adj. $p$ value = 4.13 × 10$^{-7}$) (Fig. 4g). Moreover, our SCNA analysis revealed an increase in focal gains of *EGFR* with age in low-grade glioma but not in glioblastoma (Fig. 3d), suggesting differences in the age-associated genomic landscape between the two glioma types. Together with the SCNA results, gliomas from younger patients are associated with *IDH1*, *ATRX*, and *TP53* mutations, lower SCNAs, and longer survival. In contrast, gliomas from older patients were more likely to be IDH-WT with *EGFR* mutations, chromosome 7 gain and 10 loss, *CDKN2A* deletion and worse prognosis. This clearly highlights biological differences between age-related subtypes in gliomas.

Mutations in *CDH1* were more frequent in younger stomach cancer patients (OR = 0.9414, 95% CI = 0.9027–0.9800, adj. $p$ value = 0.006), but more common in older breast cancer patients (OR = 1.0218, 95% CI = 1.0049–1.0392, adj. $p$ value = 0.0183) (Fig. 4f). This result highlights cancer-specific patterns of genomic alterations with age. We tested whether age-associated subtypes could explain differences in mutation with age. Using subtype information from a previous TCGA study[40], *CDH1* mutations were found more often in the genomically stable (GS) subtype of stomach cancer (two-sided Fisher's exact, $p$ value = 2.0 × 10$^{-5}$), which was presented more frequently in younger patients (two-sided Wilcoxon rank sum test, $p$ value = 0.0058)

(Supplementary Fig. 10a, b). As expected, *CDH1* mutations were highly enriched in the invasive lobular carcinoma (ILC) subtype of breast cancer[35] (two-sided Fisher's exact, $p$ value = 4.4 × 10$^{-38}$), which was more prevalent in older patients (two-sided Wilcoxon rank sum test, $p$ value = 0.00081) (Supplementary Fig. 10c, d). Overall, our results demonstrate that non-silent mutations in cancer-driver genes were not uniformly distributed across ages and we have comprehensively identified, based on data available at present, genes that show age-associated mutation patterns, which might partly be explained by the presence of age-related subtypes in some cancers. These patterns might point out age-associated disparities in carcinogenesis, molecular subtypes and survival outcome.

**Age-associated alterations in oncogenic signalling pathways.** As we have identified numerous age-associated alterations in cancer-driver genes both at the level of somatic mutations and SCNAs, we asked if the age-associated patterns also exist in particular oncogenic signalling pathways. We used the data from a previous TCGA study, which had comprehensively characterised 10 highly altered signalling pathways in cancers[41]. To make the subsequent analysis comparable to previous analyses, we restricted the analysis to samples that were used in our previous analyses, yielding 8055 samples across 33 cancer types (Table 1). Using logistic regression adjusting for gender, race and cancer type, we identified five out of 10 signalling pathways that showed a positive association with age (adj. $p$ value < 0.05), indicating that the genes in these pathways are altered more frequently in older patients, concordant with the increase in overall mutations and SCNAs with age (Fig. 5a, Supplementary Data 12). The strongest association was found in cell cycle (OR = 1.0122, 95% CI = 1.0076–1.0168, adj. $p$ value = 1.40 × 10$^{-6}$) and Wnt signalling (OR = 1.0122, 95% CI = 1.0073–1.0172, adj. $p$ value = 6.39 × 10$^{-6}$) pathways. We next applied logistic regression to investigate the cancer-specific association between age and oncogenic signalling alterations for cancer types that contained at least 100 samples. In total, we identified 28 significant associations across 15 cancer types (adj. $p$ value < 0.05) (Fig. 5b, Supplementary Data 12). Alterations in Hippo and TP53 signalling pathways significantly associated with age, both positively and negatively, in five cancer types. Consistent with our pan-cancer analysis, cell cycle, Notch and Wnt signalling each showed an increase in alterations with age in three cancer types. We found that alterations in cell cycle pathway increased with age in low-grade glioma (OR = 1.0313, 95% CI = 1.0161–1.0467, adj. $p$ value = 0.00035). This was largely explained by the increase in *CDKN2A* and *CDKN2B* deletions with age as well as epigenetic silencing of *CDKN2A* in older patients (Fig. 5c). On the other hand, TP53 pathway alteration was more pronounced in younger patients (OR = 0.9520, 95% CI = 0.9372–0.9670, adj. $p$ value = 2.63 × 10$^{-8}$), due to mutations in the *TP53* gene (Fig. 5c). In endometrial cancer, two pathways—Hippo (OR = 0.9681, 95% CI = 0.9459–0.9908, adj. $p$ value = 0.0126) and Wnt (OR = 0.9741, 95% CI = 0.9541–0.9946, adj. $p$ value = 0.0240)—showed a negative association with age, that may be explained by the presence of hypermutated tumours in younger patients. Collectively, we report pathway alterations in relation to age in several cancer types, highlighting differences in oncogenic pathways that might be important in cancer initiation and progression in an age-related manner.

**Age-associated gene expression and DNA methylation changes.** Apart from the genomic differences with age, we investigated age-associated transcriptomic and epigenetic changes across cancers. We separately performed multiple linear regression analyses on gene expression data and methylation data of 24 cancer types that contained at least 100 samples in both types of data (Table 1). We

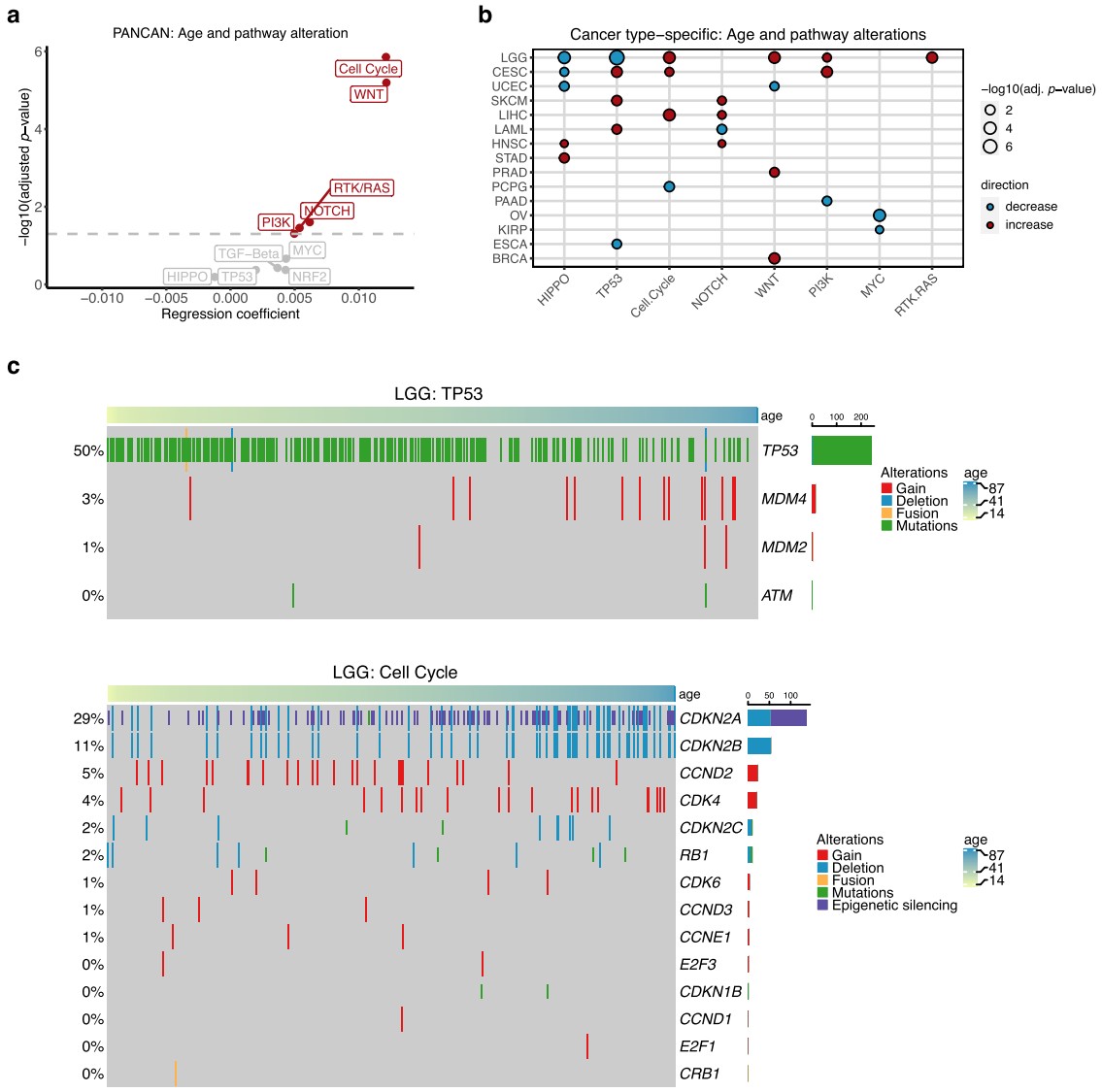

**Fig. 5 Association between cancer patients' age and oncogenic signalling pathway alterations. a** Association between age and oncogenic pathway alterations in the pan-cancer level. Multiple logistic regression coefficients and significant values are shown. Multiple-hypothesis testing correction was done using Benjamini–Hochberg procedure. Pathways with a significant positive association between age and alterations (adj. *p* value < 0.05) are highlighted in red. **b** Cancer-specific age-associated pathway alterations. Pathways that show a significant positive and negative association with age per cancer type (multiple logistic regression, adj. *p* value < 0.05) are displayed in red and blue dots, respectively. Multiple-hypothesis testing correction was done using Benjamini–Hochberg procedure. **c** Heatmap showing age-associated alterations in genes associated with TP53 and cell cycle pathways in LGG. Samples are sorted by age. Colours represent types of alteration. TCGA cancer type acronyms and their associated name are provided in Table 1.

noticed that, across all genes, the regression coefficient of age on gene expression negatively correlated with the regression coefficient of age on methylation in all cancer types (Supplementary Fig. 11), suggesting that global changes of gene expression and methylation with age are in the opposite direction. This supports the established role of DNA methylation in suppressing gene expression. Numbers of significant differentially expressed genes with age (age-DEGs) (adj. *p* value < 0.05, Supplementary Data 13) varied from nearly 5000 up- and down-regulated genes in low-grade glioma to no significant gene in 5 cancer types. Similarly, we also identified significant differentially methylated genes with age (age-DMGs, Supplementary Data 14) (adj. *p* value < 0.05), and the numbers of age-DEGs and age-DMGs were consistent for most cancer types (Fig. 6a). It is worth noting that cancers of female reproductive organs, including breast, ovarian and endometrial cancers show among the highest number of age-DEGs and age-DMGs.

To exclude the possibility that germline predisposition mutations in some patients may cause such a high number of age-DEGs and age-DMGs in cancers of the female reproductive system, we excluded samples harbouring germline mutations in *BRCA1*, *BRCA2* and *TP53* as previously identified[42] from the breast, ovarian and endometrial cancer cohorts and repeated the multiple linear regression analysis. We observed a high correlation between regression coefficients of the analyses from all tumours and the analyses excluding samples with germline mutations, for all three cancer types and for both gene expression and methylation ($R = 0.93–0.99$, *p* value < $2.2 \times 10^{-16}$) (Supplementary Fig. 12). The overlap between age-DEGs or age-DMGs identified from all samples and from samples without germline variants were large (Supplementary Fig. 12, Supplementary Data 15). Therefore, the high number of age-DEGs and age-DMGs are independent of the presence of germline predisposition mutations in some patients. We, therefore, used age-DEGs

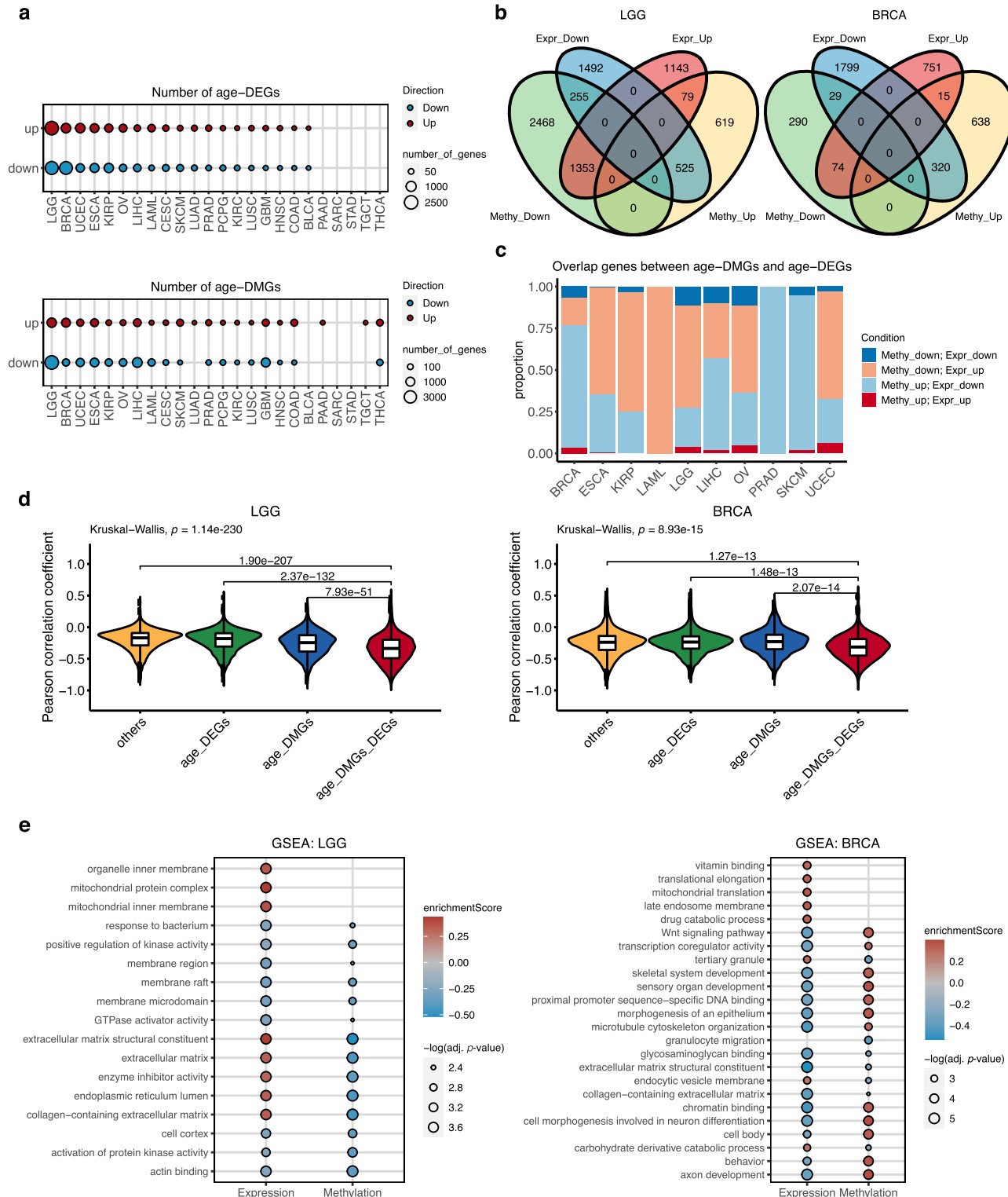

and age-DMGs identified from all samples for subsequent analyses.

We next focused our analysis on ten cancer types that contained at least 150 age-DEGs and 150 age-DMGs, including low-grade glioma, breast cancer, endometrial cancer, oesophageal cancer, papillary renal cell carcinoma, ovarian cancer, liver cancer, acute myeloid leukaemia, melanoma, and prostate cancer. We identified overlapping genes between age-DEGs and age-DMGs and found that most of them, from 84% (37/44 genes) in ovarian cancer to 100% in acute myeloid leukaemia (57 genes)

and prostate cancer (7 genes), were genes that presented increased methylation and decreased expression with age and genes that had decreased methylation and increased expression with age (Fig. 6b, c, Supplementary Fig. 13, Supplementary Data 16). We further examined the correlation coefficient between methylation and expression comparing between 4 groups of genes 1) genes overlapping between age-DMGs and age-DEGs (age-DMGs-DEGs), 2) age-DMGs only, 3) age-DEGs only, and 4) other genes. We found that age-DMGs-DEGs had the most negative correlation between DNA methylation and

**Fig. 6 Age-related gene expression in cancers was controlled by age-related methylation. a** Number of age-DEGs and age-DMGs across cancer types. Red dots represent up-regulated genes, while blue dots denote down-regulated genes. The dot size corresponds to the number of genes. **b** Venn diagrams of the overlap between age-DEGs and age-DMGs. LGG and BRCA are shown as examples. Venn diagrams of the other cancers are shown in Supplementary Fig. 13. **c** The distribution of overlap genes between age-DMGs and age-DEGs. The genes were classified into (1) down-regulated methylation and down-regulated expression, (2) down-regulated methylation and up-regulated expression, (3) up-regulated methylation and down-regulated expression, and (4) up-regulated methylation and up-regulated expression. **d** Violin plots showing the distribution of the Pearson correlation coefficient between methylation and gene expression in LGG and BRCA. Genes were grouped into (1) common genes between age-DMGs and age-DEGs (age-DMGs-DEGs), (2) age-DMGs only genes, (3) age-DEGs only genes, and (4) other genes. LGG others $n = 5841$, age_DEGs $n = 2635$, age_DMGs $n = 3087$, age_DMGs_DEGs $n = 2212$ genes; BRCA others $n = 9726$, age_DEGs $n = 2550$, age_DMGs $n = 928$, age_DMGs_DEGs $n = 438$ genes. The group comparison was performed by the Kruskal–Wallis test. The pairwise comparisons were done using two-sided Dunn's test. P values from Dunn's test between age-DMGs-DEGs and the other groups adjusted by Bonforroni correction are shown. The plots for the other cancers are shown in Supplementary Fig. 14. **e** The enriched gene ontology (GO) terms identified by Gene set enrichment analysis (GSEA) in LGG and BRCA. The dot size corresponds to a significant level (permutation test). Multiple-hypothesis testing correction was done using Benjamini–Hochberg procedure. A GO term was considered significantly enriched term if adj. *p* value < 0.05 for gene expression and adj. *p* value < 0.1 for methylation. Colours represent enrichment scores, red denotes positive score (enriched in older patients), while blue signifies negative score (enriched in younger patients). The plots for the other cancers are shown in Supplementary Fig. 15. TCGA cancer type acronyms and their associated name are provided in Table 1.

expression when comparing with other groups of genes (Fig. 6d, Supplementary Fig. 14, Supplementary Data 17), highlighting that age-associated gene expression changes in cancer are repressed, at least in part, by DNA methylation.

We next performed Gene Set Enrichment Analysis (GSEA) to gain biological insights into expression and methylation changes with age. We identified various significantly enriched Gene Ontology (GO) terms across cancers (adj. *p* value < 0.05 for gene expression and adj. *p* value < 0.1 for methylation) (Fig. 6e, Supplementary Fig. 15, Supplementary Data 18). Notably, several GO terms were enriched in both expression and methylation changes, in the opposite direction. The enriched terms in breast cancer included several signalling, metabolism, and developmental pathways. The Wnt signalling pathway, which was altered more frequently in older breast cancer patients (Fig. 5b), showed a decrease in gene expression and increase in methylation with age. In low-grade glioma, interestingly, mitochondrial terms were enriched in the gene expression of older patients. Mitochondrial dysfunction is known to be important in glioma pathophysiology[43], thus the different levels of mitochondrial aberrations might contribute to disparities in the aggressiveness of gliomas in patients of different age. We also identified numerous immune-related terms enriched across several cancer types, including oesophageal, papillary renal cell, liver, and prostate cancers (Supplementary Fig. 15, Supplementary Data 18). Recent studies suggested alterations in immune-related gene expression and immune cell abundance changes with age in cancers[44–46]. In the present study, we have systematically characterised the transcriptome and methylation in relation to age across cancer types. Our results suggest that gene expression changes with age in cancer are controlled, at least in part, by DNA methylation. These changes reflect differences in biological pathways that might be important in tumour development.

## Discussion

Although age is an important risk factor for cancer, how age impacts the molecular landscape of cancer is not well understood. In this study, we provide a comprehensive multi-omics overview of the age-associated molecular landscape in cancer, including GI, LOH, WGD, SCNAs, somatic mutations, pathway alterations, gene expression and DNA methylation. We confirmed the known increase in mutation load[4,5], that can be in part explained by an increase in C>T mutations, and found an increase in GI, LOH and WGD with age in several cancer types. We identified several age-related pan-cancer and cancer-specific alterations. The highest age-related differences were evident in low-grade glioma and endometrial cancer.

Cancer develops through the accumulation of genetic and epigenetic alterations. Mutation accumulation with age is thought to be a cause of cancer and a substantial portion of mutations arise before cancer initiation[6]. The age-associated mutation accumulation has been demonstrated in both cancer[4,5] and normal tissues[47–49], providing a better understanding of early carcinogenic events. Our results show that, in addition to mutations, SCNAs, LOH and WGD increase with age in several cancers, in particular low-grade glioma, endometrial and ovarian cancers. Recent evidence suggests that SCNA burden is a prognostic factor associated with both recurrence and death[50], thus, an increased SCNA level with age might relate to poor prognosis in the elderly.

The negative association between age and mutation in *IDH1*, *ATRX* and *TP53* in glioma points towards the difference of patient age at diagnosis between the *IDH*-mutant and *IDH*-WT subtypes. *IDH*-mutant tumours are observed in the majority of low-grade glioma and show favourable prognosis. *IDH*-WT low-grade gliomas, on the other hand, more resemble glioblastomas and have poorer survival. In glioblastoma, although *IDH*-mutant cases are a minority of tumours, they are also associated with younger age[51]. A recent functional study in neural stem cells (NSCs) showed that the combination of *IDH1*, *ATRX* and *TP53* alterations blocks NSC differentiation by causing hypermethylation of CTCF motifs flanking the *SOX2* locus, disrupting chromatin looping and dysregulation of *SOX2*, an important transcription factor in self-renewal and differentiation of NSCs[52]. Impaired differentiation, growth arrest evasion by mutations in *TP53*, and alternative lengthening of telomeres by *ATRX* inactivation thus cooperatively promote gliomas in younger patients. The present study together with others[38,53], therefore indicates that glioma shows unique age-associated subtypes. However, more research is needed to understand how age influences the evolution of glioma subtypes.

Our results highlighted substantial age-associated differences in the genome of endometrial cancer. Younger endometrial tumours associate with a *POLE* and MSI-H subtypes, leading to an enrichment of hypermutated tumours, while tumours from older patients tend to harbour more SCNAs and lower mutation load. Previous studies have classified endometrial cancer into four subtypes: *POLE*, MSI-H, copy-number low and copy-number high subtypes. The *POLE* subtype and MSI-H subtype are dominated by *POLE* and defective mismatch repair mutational signatures, respectively[36]. Conversely, the copy-number low and copy-number high subtypes had a dominant ageing-related mutational signature[34]. The *POLE* and MSI-H subtypes have a favourable prognosis, while the copy-number high subtype is

associated with poor survival. Therefore, endometrial cancer from younger patients is associated with *POLE* mutations, mismatch repair defects, high mutation load and better survival outcomes. Older endometrial cancer, however, is related to extensive SCNAs and worse prognosis. Importantly, apart from low-grade glioma and endometrial cancer, we demonstrate that other cancer types also present an age-associated genomic landscape in cancer-driver genes and oncogenic signalling pathways. Indeed, some of these age-related differences might be explained by age-related subtypes, such as the high prevalence of *CDH1* mutation in invasive lobular breast carcinoma and GS stomach cancer, that are presented more often in older and younger patients, respectively. Further detailed investigation in each cancer type is required to fully distinguish age-related subtype effects from age-associated effects, or even to identify a new age-related subtype. Our perspective is that whether age-associated genomic differences come from age-related subtypes or from age-related features they reflect biological differences between cancers at different ages, highlighting the impact of age on the molecular profile of cancer. One limitation, however, of our age-related genomic analyses is that we did not include in our model information on quantitative differences of genomic alterations, such as homozygous/heterozygous loss and clonal/subclonal mutations, to not over-complicate the analyses. Furthermore, how and why age-associated genomic differences and age-related subtypes occur still remain to be studied.

Having identified age-related differences in the molecular landscapes of various cancers, the obvious question is: what drives these differences? Accumulating evidence has underscored the importance of tissue environment changes with ageing in cancer initiation and progression[7,8,45,54]. We reason that tissue environment changes during ageing might provide different selective advantages for tumours harbouring different molecular alterations, in turn directing tumours to different evolutionary routes. Therefore, cancers with different genomic alterations might thrive better in younger or older patients. Gene expression and epigenetic changes related to ageing have been studied and linked to cancer[8,44,55,56]. Here, we identified numerous age-associated gene expression and corresponding DNA methylation changes in a broad range of cancers. Indeed, age-DMGs-DEGs are those with the strongest negative correlation between methylation and expression when comparing with other groups, indicating that differentially expressed genes with age in cancer are partly regulated by methylation. Expression and methylation changes with age link to several biological processes, showing that cancer from patients with different ages present different phenotypes. We also noticed that cancer in female reproductive organs including breast, ovarian and endometrial cancers are among those with the highest number of age-DEGs and age-DMGs. These cancers tend to have a higher mass-normalised cancer incidence, which may reflect evolutionary trade-offs involving selective pressures related to reproduction[57]. Age-associated hormonal changes could also be responsible for this age-related expression differences in cancer, as evidenced by studies in breast cancer[58,59]. A limitation of this analysis is that we chose only one methylation probe per gene to create a one-to-one mapping between genes and probes. Other probes might also have an impact on gene expression as well as might cause noise in our GSEA analysis from methylation data. Next, although we have already included tumour purity in our linear model, it is not possible to account for the different tumour-constituent cell proportions and thus fully exclude the influence of gene expression in non-cancerous cells such as infiltrating immune cells[45]. Further studies are required to provide mechanistic understanding of the impact of an ageing microenvironment in shaping tumour evolution.

During the preparation of our manuscript, a study based on a similar concept has been released by Li et al.[60]. In this work, Li et al. used TCGA and the recent pan-cancer analysis of whole genomes data to study age-associated genomic differences in cancer. Results from the two studies are consistent in several respects. Firstly, both studies indicate an increase in mutations and SCNAs as a function of age. In addition, despite using slightly different statistical cutoffs and models, several age-associated genomic features are identified by both studies, for example, the higher frequency of *IDH1* and *ATRX* mutations in younger glioma patients. Li et al. explored mutational timing and signatures, which suggested possible underlying mechanisms for age-associated genomic differences. Our study, however, has also featured an age-related genomic profile in endometrial cancer. We have investigated cancer-specific associations between age and LOH, WGD and oncogenic signalling. Furthermore, we have analysed age-related global transcriptomic and DNA methylation changes. Both complementary studies thus serve as a foundation for understanding age-related differences and effects on the cancer molecular landscape and emphasise the importance of age in cancer genomic research that is particularly valuable in clinical practice.

## Methods

**Data acquisition.** Publicly available copy-number alteration seg files (nocnv_hg19.seg), normalised mRNA expression in RSEM (.rsem.genes.normalized_results TCGA files from the legacy archive, aligned to hg19), and clinical data (XML files) from TCGA were downloaded using *TCGAbiolinks* (version 2.14.1)[61]. The mutation annotation format (MAF) file was downloaded from the TCGA MC3 project[62] (https://gdc.cancer.gov/about-data/publications/mc3-2017). The somatic alterations in 10 canonical oncogenic pathways across TCGA samples were obtained from a previous study by Sanchez-Vega et al.[41]. The TCGA Illumina Human-Methylation450K array data (in β values) was downloaded from Broad GDAC Firehose (http://gdac.broadinstitute.org/). The allele-specific copy-number, tumour ploidy, tumour purity that were estimated using ASCAT (version 2.4.2)[63] on hg19 SNP6 arrays with penalty = 70 were obtained from previous studies[64,65], available at (https://github.com/Crick-CancerGenomics/ascat/tree/master/ReleasedData/TCGA_SNP6_hg19). We restricted our subsequent analyses to samples that have these profiles available. WGD duplication was determined using fraction of genome with LOH and ploidy information. GI scores have been computed as fraction of genomic regions that are not in 1 + 1 (for non WGD tumours) or 2 + 2 (for WGD tumours) statuses. For each data type and each cancer type, the summary of the numbers of TCGA samples included in the analysis, alongside clinical variable analysed are presented in the Supplementary Data 1.

**Statistical analysis and visualisation.** Simple linear regression and multiple linear regression adjusting for clinical variables were performed using the *lm* function in R to access the relationship between age and continuous variables of interest. Simple logistic regression to investigate the association between age and binary response (e.g. mutation as 1 and wild-type as 0) and multiple logistic regression adjusting for covariates were carried out using the *glm* function in R. In pan-cancer analyses, gender, race and cancer type were variables included in the linear model. Clinical variables used in cancer-specific analyses included gender, race, pathologic stage, neoplasm histologic grade, smoking status, alcohol consumption and cancer-specific variables such as oestrogen receptor (ER) status in breast cancer. To avoid the potential detrimental effect caused by missing data, we retained only variables with missing data less than 10% of samples used in the somatic copy-number alteration analysis (Supplementary Data 1). To account for the difference in the proportion of cancer cells in each tumour, tumour purity (cancer cell fraction) estimated from ASCAT was included in the linear model. When necessary, to avoid the separation problem that might occur due to the sparse-data bias[66], *logistf* function from the *logistf* package (version 1.23)[67] was used to perform multivariable logistic regression with Firth's penalisation[68]. Effect sizes from logistic regression analyses were reported as odds ratio per year and 95% confidence intervals. P values from the analyses were accounted for multiple-hypothesis testing using Benjamini–Hochberg procedure[69]. Statistical significance was considered if adj. $p$ value < 0.05, unless specifically indicated otherwise. All statistical analyses were carried out using R (version 3.6.3)[70]. Plots were generated using *ggplot2* (version 3.3.2)[71], *ggrepel* (version 0.8.2)[72], *ggpubr* (version 0.4.0)[73], *ComplexHeatmap* (version 2.2.0)[74], and *VennDiagram* (version 1.6.20)[75].

**GI score analysis.** GI score was calculated as a genome fraction (percent-based) that does not fit the estimated tumour ploidy, 2 for normal diploid, and 4 for tumours that have undergone the WGD process. Simple linear regression was

performed to identify the association between age and GI score. For pan-cancer analysis, multiple linear regression was used to adjust for gender, race, and cancer type. For cancer-specific analysis, multiple linear regression accounting for clinical variables was conducted on the cancer types that had a significant association between age and GI score from the simple linear regression analysis (adj. *p* value < 0.05). The complete set of results is presented in Supplementary Data 2.

**Percentage genomic LOH quantification and analysis**. To quantify the percent genomic LOH for each tumour, we used allele-specific copy-number profiles from ASCAT. X and Y chromosome regions were discarded from the analysis. The LOH segments were segments that harbour only one allele. The percent genomic LOH was defined as 100 times the total length of LOH regions/length of the genome.

Simple linear regression and multiple linear regression adjusting for gender, race, and cancer types were conducted to investigate the relationship between age and the percent genomic LOH in the pan-cancer analysis. For cancer-specific analysis, simple linear regression was performed followed by multiple linear regression accounting for clinical factors for cancers with a significant association in simple linear regression analysis (adj. *p* value < 0.05). The complete set of results is in Supplementary Data 3.

**WGD analysis**. WGD status for each tumour was obtained from fraction of genome with LOH and tumour ploidy. To investigate the association between age and WGD across the pan-cancer dataset, we performed simple logistic regression and multiple logistic regression correcting for gender, race, and cancer type. For cancer-specific analysis, simple logistic regression was performed to access the association between age and WGD on tumours from each cancer type. Cancer types with a significant association between age and WGD (adj. *p* value < 0.05) were further subjected to the multiple logistic regression accounting for the clinical variables. The complete set of results is in Supplementary Data 4.

**List of known cancer-driver genes**. We compiled a list of known cancer-driver genes from (1) the list of 243 COSMIC classic genes from COSMIC database version 91[25] (downloaded on 1st July 2020), (2) the list of 260 significantly mutated genes from Lawrence et al.[27] and (3) the list of 299 cancer-driver genes from the TCGA Pan-Cancer study[26]. In total, we obtained 505 cancer genes and focused on the mutations and focal-level SCNAs on these genes in our study. The full list of cancer-driver genes is available in Supplementary Data 8.

**Recurrent SCNA analysis**. Recurrent arm-level and focal-level SCNAs of each cancer type were identified using GISTIC2.0[22]. Segmented files (nocnv_hg19.seg) from TCGA, marker file and CNV file, provided by GISTIC2.0, were used as input files. The parameters were set as follows: '-genegistic 1 -smallmem 1 -qvt 0.25 -ta 0.25 -td 0.25 -broad 1 -brlen 0.7 -conf 0.95 -armpeel 1 -savegene 1'. Based on these parameters, broad events were defined as the alterations happen in more than 70% of an arm. The log2 ratio thresholds for copy-number gains and losses were 0.25 and −0.25, respectively. The confidence level was set as 0.95 and the *q* value was 0.25.

To investigate the association between age and arm-level SCNAs for each cancer type, simple logistic regression was performed for each chromosomal arm that was identified as recurrent SCNA in a cancer type. Only cancer types with more than 100 samples were included in this analysis (Table 1). Arms with a significant association (adj. *p* value < 0.05) were further adjusted for clinical variables using multiple logistic regression. The complete set of results is in Supplementary Data 6. Similarly, simple and multiple logistic regression was conducted on the focal-level SCNAs for each cancer type. Regions that are overlapped with centromeres or telomeres were removed from the analysis. The complete set of results is in Supplementary Data 7.

To confirm the impact of SCNAs on gene expression, we investigated the correlation between GISTIC2.0 score and RNA-seq based gene expression (log2 (normalised RSEM + 1)) for tumours that have both types of data using Pearson correlation. The correlation was considered significant if the *p* value corrected for multiple-hypothesis testing using the Benjamini–Hochberg procedure < 0.05. The complete set of results is in Supplementary Data 7.

**SCNA score quantification and analysis**. Previous studies have developed the SCNA score representing the SCNA level of a tumour[12,23]. We applied the methods described by Yuan et al.[12] to calculate SCNA scores. Using SCNA profiles from GISTIC2.0 analysis, SCNA scores for each tumour were derived at three different levels (chromosome-, arm-, and focal-level). For each tumour, each focal-event log2 copy-number ratio from GISTIC2.0 was classified into the following score: 2 if the log2 ratio ≥ 1, 1 if the log2 ratio < 1 and ≥0.25, 0 if the log2 ratio < 0.25 and ≥−0.25, −1 if the log2 ratio < −0.25 and ≥−1, and −2 if the log2 ratio < −1. The |score| from each focal event in a tumour was then summed into a focal core of a tumour. Thereafter, the rank-based normalisation (rank/number of tumours in a cancer type) was applied to focal scores from all tumours within the same cancer type, resulting in normalised focal-level SCNA scores. Therefore, tumours with high focal-level SCNAs will have focal-level SCNA scores close to 1, while tumours with low focal-level SCNAs will have scores close to 0. For the arm- and chromosome-level SCNA scores, a similar procedure was applied to the broad

event log2 copy-number ratio from GISTIC2.0. An event was considered as a chromosome-level if both arms have the same log2 ratio, otherwise it was considered as an arm-level. Similar to the focal-level SCNA score, each arm- and chromosome-event log2 copy-number ratio was classified into the 2, 1, 0, −1, −2 scores using the threshold described above. The |score| from all arm-events and chromosome-events for a tumour were then summed into an arm score and chromosome score, respectively. For each cancer type, the rank-based normalisation was applied to arm scores and chromosome scores from all tumours to derive normalised arm-level SCNA scores and normalised chromosome-level SCNA scores, respectively. An overall SCNA score for a tumour was defined as the sum of focal-level, arm-level, and chromosome-level SCNA scores. A chromosome/arm-level SCNA score for a tumour was defined as the sum of chromosome-level and arm-level SCNA scores.

The association between age and overall, chromosome/arm-level, and focal-level SCNA scores for each cancer type was investigated using simple linear regression. Cancer types with a significant association (adj. *p* value < 0.05) were then subjected to multiple linear regression analysis adjusting for the clinical variables. The complete set of results is included in Supplementary Data 5.

**Analysis of age-associated somatic mutation in cancer genes**. We obtained the mutation data from the MAF file from the recent TCGA Multi-Centre Mutation Calling in Multiple Cancers (MC3) project[62]. In the MC3 effort, variants were called using seven variant callers. We filtered the variants to keep only non-silent SNVs and indels located in gene bodies, retaining only 'Frame_Shift_Del', 'Frame_Shift_Ins', 'In_Frame_Del', 'In_Frame_Ins', 'Missense_Mutation', 'Nonsense_Mutation', 'Nonstop_Mutation', 'Splice_Site' and Translation_Start_Site in the 'Variant_Classification' column. We focused only on mutations in the cancer genes from our compiled list of cancer-driver genes. To prevent the bias that might cause by hypermutated tumours, we restricted the analysis to tumours with <1000 mutations per exome. For pan-cancer analysis, multiple logistic regression accounting for gender, race and cancer type was performed to investigate the association between age and mutations in 20 cancer genes that are mutated in >5% of samples (Supplementary Data 11). For cancer-specific analysis, simple logistic regression was used to identify cancer genes that the mutations in these genes are associated with the patient's age. Only genes that are mutated in >5% of samples from each cancer type were included in the analysis. The significant associations (adj. *p* value < 0.05) were further investigated using multiple logistic regression accounting for clinical variables. The complete set of results is in Supplementary Data 11.

**Analysis of mutation burden, substitution classes, MSI-H status, and POLE/POLD1 mutations**. A mutation burden was defined as the total non-silent mutations in an exome. A package *maftools* (version 3.3.2) was used to import and extract information from maf files[76]. The mutation burden for each tumour was log-transformed before using it in the subsequent analysis. To investigate the relationship between age and mutation burden in pan-cancer, multiple linear regression adjusting for gender, race and cancer type was conducted. For cancer-specific analysis, simple linear regression was performed. Cancer types with a significant association between age and mutation burden in simple linear regression analysis (adj. *p* value < 0.05) were further examined using multiple linear regression accounting for clinical factors. The complete set of results is in Supplementary Data 9. Similarly, multiple linear regression adjusting for gender, race and cancer type was used to examine the relationship between age and each of the six substitution classes in pan-cancer. For cancer-specific analysis, simple linear regression was performed. Cancer types with a significant association between age and fraction contribution of a substitution class in simple linear regression analysis (adj. *p* value < 0.05) were further examined using multiple linear regression accounting for clinical factors. The complete set of results is in Supplementary Data 10.

MSI status for COAD, READ, STAD, and UCEC were downloaded from TCGA using *TCGAbiolinks*. To study the association between the presence of high microsatellite instability (MSI-H) and age, tumours were divided into binary groups: MSI-H = TRUE and MSI-H = FALSE. Multiple logistic regression adjusting for clinical variables was then performed. Similarly, *POLE* and *POLD1* mutation status were in a binary outcome (mutated and not mutated). Multiple logistic regression was used to investigate the association between age and *POLE/POLD1* mutations in cancer types that contained POLE/POLD1 mutations in >5% of samples.

**Oncogenic signalling pathway analysis**. We used the list of pathway-level alterations in ten oncogenic pathways (cell cycle, Hippo, Myc, Notch, Nrf2, PI-3-Kinase/Akt, RTK-RAS, TGFβ signalling, p53 and β-catenin/Wnt) for TCGA tumours comprehensively compiled by Sanchez-Vega et al.[41]. Member genes in the pathways were accessed for SCNAs, mutations, epigenetic silencing through promoter DNA hypermethylation and gene fusions. We retained only the pathway alteration data of samples that were presented in our SCNA analysis. For the pan-cancer analysis, we employed multiple logistic regression adjusting for the patient's gender, race and cancer type to demonstrate the relationship between pathway-level alteration and age. To investigate the association between age and cancer-

specific pathway alterations, we performed simple logistic regression. Cancer types with a significant association (adj. p value < 0.05) were further examined by multiple logistic regression accounting for clinical variables. The complete set of results is in Supplementary Data 12.

**Gene expression and DNA methylation analysis**. To render the results from gene expression and DNA methylation comparable, we limited the analysis to genes that are presented in both types of data. The lowly expressed genes were filtered out from the analysis by keeping only genes with RSEM > 0 in more than 50 percent of samples. Only protein coding genes identified using biomaRt[77] (version 2.46.0, data based on Ensembl version 100, April 2020) were included in the analyses. Normalised mRNA expression in RSEM for each TCGA cancer type was log2-transformed before subjected to the multiple linear regression analysis adjusting for clinical factors. RNA-seq data for colon cancer and endometrial cancer consisted of two platforms, Illumina HiSeq and Illumina GA. Thus, a platform was included as another covariate in the linear regression model for these two cancer types. Genes with adj. p value < 0.05 were considered significantly differentially expressed genes with age (age-DEGs) (Supplementary Data 13). DNA methylation data was presented as β values, which are the ratio of the intensities of methylated and unmethylated alleles. Because multiple methylation probes can be mapped to the same gene, we used the one-to-one mapping genes and probes by selecting the probes that are most negatively correlated with the corresponding gene expression from the meth.by_min_expr_corr.data.txt files downloaded from Broad GDAC Firehose. Similar multiple linear regression to the gene expression analysis was performed on the methylation data. Genes with adj. p value < 0.05 were considered significant differentially methylated genes with age (age-DMGs). The complete set of results is in Supplementary Data 14. To exclude the possibility that germline predisposition mutations in some patients may cause this high number of age-DEGs and age-DMGs in female reproductive cancers, we excluded samples harbouring germline mutations in *BRCA1*, *BRCA2* and *TP53* as previously identified[42] from breast, ovarian and endometrial cancer cohorts. In total, we removed ~13% of patients from breast and endometrial cancer and ~20% of patients from ovarian cancer. After excluding these samples, we performed a similar multiple linear regression analysis. The complete set of results is in Supplementary Data 15.

The correlation between gene expression and DNA methylation was calculated using Pearson correlation. We used the Kruskal–Wallis test to investigate the differences between correlation coefficients among groups (age-DMGs-DEGs, age-DMGs, age-DEGs, other genes). The pairwise comparisons were carried out by two-sided Dunn's test. The complete set of results is in Supplementary Data 17.

GSEA was performed to investigate the GO terms that are enriched in tumours from younger or older patients. The analysis was done using the package *ClusterProfiler* (version 3.14.3)[78]. Briefly, genes or methylation probes were ranked based on their regression coefficient with age from the most positive regression coefficient with age (most up-regulated in older patients) to the gene with the most negative regression coefficient with age (most up-regulated in younger patients). A ranked gene list consisting of all genes was used in the GSEA to determine whether genes in a set of interest from GO are randomly distributed throughout the ranked gene list or are found more often in the top or the bottom of the list[79]. The complete list of enriched GO terms is presented in Supplementary Data 18.

**Reporting summary**. Further information on research design is available in the Nature Research Reporting Summary linked to this article.

## Data availability
TCGA data used in this study are publicly available and can be obtained from NCI's Genomic Data Commons portal [https://portal.gdc.cancer.gov/], TCGAbiolinks (version 2.14.1)[61] and Broad GDAC Firehose [http://gdac.broadinstitute.org/]. The mutation annotation format (MAF) file was downloaded from the TCGA MC3 project [https://gdc.cancer.gov/about-data/publications/mc3-2017]. List of known cancer-driver genes were compiled from COSMIC database[25] version 91 [https://cancer.sanger.ac.uk/cosmic], Lawrence et al.[27]. [https://doi.org/10.1038/nature12912] and TCGA Pan-Cancer study[26] [https://doi.org/10.1016/j.cell.2018.02.060]. Oncogenic signalling pathway data was obtained from Sanchez-Vega et al.[41]. [https://doi.org/10.1016/j.cell.2018.03.035]. Allele-specific copy-number, tumour ploidy, tumour purity, GI scores and WGD status of TCGA tumours generated by ASCAT (version 2.4.2) were obtained from Martincorena et al.[64] available at [https://github.com/Crick-CancerGenomics/ascat/tree/master/ReleasedData/TCGA_SNP6_hg19]. The remaining data are available within the Article, Supplementary Information or available from the authors upon request. Source data are provided with this paper.

## Code availability
The custom scripts for data analysis and generate figures are available at [https://github.com/maglab/Age-associated_cancer_genome][80] and released at [https://doi.org/10.5281/zenodo.4564690].

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

## Acknowledgements

The results published here are in whole based upon data generated by the TCGA Research Network: https://www.cancer.gov/tcga. K.C. is supported by a Mahidol-Liverpool Ph.D. scholarship from Mahidol University, Thailand, and the University of Liverpool, UK. J.P.M. is grateful to funding from the Wellcome Trust (208375/Z/17/Z) and the Biotechnology and Biological Sciences Research Council (BB/R014949/1). This work was supported by the Francis Crick Institute, which receives its core funding from Cancer Research UK (FC001202), the UK Medical Research Council (FC001202), and the Wellcome Trust (FC001202). P.V.L. is a Winton Group Leader in recognition of the Winton Charitable Foundation's support towards the establishment of The Francis Crick Institute. We wish to thank members of the Integrative Genomics of Ageing Group for suggestions and discussion. We appreciate helpful suggestions and discussion from Paul C. Boutros, Constance H. Li and members of the Boutros lab.

## Author contributions

K.C., T.L., P.V.L. and J.P.M. conceived the project and designed the study. T.L. and P.V.L. provided data. K.C. performed the analyses with help from T.L. T.L., L.P., P.V.L. and J.P.M. provided critical insights and were involved in data interpretation. K.C. wrote the first draft of the paper. All authors edited and approved the paper.

## Competing interests

The authors declare no competing interests.
