## [Peer Review File · Nature Communications]

REVIEWER COMMENTS

Reviewer #1 (Remarks to the Author): Expert in ageing and cancer

Chatsirisupachai and colleagues present integrated analyses of TCGA data to describe how genomic, epigenetic and gene expression changes in cancers are dependent on the age of the patient. Mostly, they performed linear regression analyses on various genetic, epigenetic and gene expression parameters across all cancers or individual cancers as a function of age. I believe that such a method has advantages over binning (e.g. segregating cases into halves, quartiles, etc), as one would expect progressive changes that would mirror other progressive changes associated with aging.

Genomic instability (GI) indices and LOH show age-dependent increases in multiple cancers, although it's interesting that some cancer types show the opposite trend (such as with LOH and lung AC). Similarly, increases in whole genome duplication (WGD) with age was observed across cancer types, with significant increases for some cancers like endometrial and ovarian. Notably, endometrial cancers show increases in GI, LOH and WGD with age. Consistent with the observations for GI, somatic copy number alterations (SCNA) also increased with age, evident in similar cancers as observed for GI (plus some additional ones). They also identified the chromosomal arms that tend to be gained and lost more often with age, as well as focal SCNA that were altered with age (again, endometrial cancers, ovarian cancers and low-grade gliomas stand out). Again, it is interesting that lung AC shows the opposite pattern, with decreased SCNA in cancers from older individuals. Cancers with SCNAs altered by age also exhibited corresponding alterations in the expression of known cancer driver genes.

Not surprisingly, all cancers combined and 18 individual cancers exhibit a significant increase in point mutational burden with age, likely due to the increased spontaneous deamination driven C>T changes that characterize aging tissues. The only exception was endometrial cancers, and they showed that younger endometrial cancers are more frequently MSI-H, and have more frequent POLE and POLD1 mutations (hypermutators). They also identify particular genes where mutation frequency is impacted by age (excluding hypermutator cancers); for example, IDH1 mutations in gliomas negatively correlate with age. It's interesting that mutations in the same gene can negatively and positively correlate with age for different cancers. These analyses highlight how age-dependent changes in selective pressures could interface with mutational changes to impact the mutational landscape of cancers.

They then examined age-associated changes in oncogenic signaling pathways in cancers. 5 of 10 signaling pathways that were previously shown to be deregulated in cancers showed age dependence increases in these pathways across cancers (more frequent mutational alterations or SCNA in cancers with increasing age). They extended these analyses, showing pathway alterations across 15 cancer types. Thus, selection for mutational deregulation of different pathways appears to be differential across various cancers as a function of age.

For some cancers like gliomas, a large number of genes show differential expression dependent on age, while for other cancers there were no such genes. Clearly, some cancers like gliomas exhibit

very different biology dependent on the age of the host, while others less so. Changes in gene expression were negatively concordant with those for DNA methylation, as would be expected, although the strong correlations observed suggest a major role for such DNA methylation in controlling age-dependent changes in gene expression in cancers. They observed similar negative correlations between gene expression and DNA methylation for some Gene Ontology terms.

In all, these bioinformatic analyses of TCGA data to determine the age-dependence of mutational, epigenetic and gene expression changes in cancers is thorough and valuable. The assumptions made and other considerations (such as the use of cutoffs in terms of case #s for some analyses) appear well-justified, and their interpretations of the data appear reasonable. The Intro and Discussion both define the problems and offer a good analysis of the implications of their results (such as to why female reproductive cancers show such clear age-dependence in epigenetic parameters). The Methods are clear and seem appropriate (although genomic bioinformatics is not my area of expertise). Obviously, a comprehensive analysis of genetic and expression changes dependent on age across cancers is going to raise more questions than it answers, providing fodder for hypothesis testing in the years to come. And that's what makes this study very useful.

These analyses will serve as a valuable resource for investigators interested in associations between age and cancers. They also highlight the important role that age plays in sculpting the mutational and phenotypic landscapes of cancers, influenced both by selection and mutational processes. This study clearly shows that the age of the patient matters when it comes to the biology of the cancer, which should have profound implications for how we develop preventative, diagnostic and therapeutic interventions.

Finally, as they cite, a study deposited on bioRxiv by Li et al used TCGA and PCAWG databases to perform a similar analysis for how aging impacts cancer genomics. The current study and this study find some common patterns, but due to their different methods also make unique discoveries (for example, related to the global transcriptomic and DNA methylation changes revealed from the current study). In the end, the field will be well served by the availability of both studies, leading to stronger conclusions overall and highlighting some unique contributions from each study.

Minor:

- 1) Suppl Fig 5A, showing the increased point mutational burden with age of cancer (all combined), should be included in a main figure (like Fig 4). I imagine that the authors chose to put it supplemental given that this association is known, but if it can be squeezed into to a main figure, it will help the reader better appreciate the full picture of mutational changes with age.
- 2) The GI score is very well described in the Methods, but it would be helpful if the authors very briefly described the basics of this score in the Results section when these analyses are presented.

Signed: James DeGregori

Reviewer #2 (Remarks to the Author): Expert in cancer genomics

In the current study, Chatsirisupachai et al. reported an integrative analysis of TCGA data to identify age-associated patterns of somatic alterations in human cancers. This is an important research question given that aging is the #1 risk factor of cancer. The analysis included multiple data types such as copy number, LOH, somatic mutations, expression, and methylation. Overall, this is a very large multi-omics study that provides novel insights into cancer and aging. The findings are scientifically interesting and clinically relevant. However, there are also some technical questions, many of which are related to the complexity of the disease, that could be better addressed.

Major:

1. For the observed age-related genomic differences that are specific to certain cancers, for instances, the more frequent hypermutators in younger endometrial cancer patients, and IDH-WT vs IDH-mutant in glioma, do they represent unique features of aging, or age-related subtypes of certain cancer? Either could be important, but may need to be distinguished or discussed carefully. The latter could be explained by a heterogeneous nature of the disease itself that may or may not have been discovered. Similarly, for somatic mutations, certain cancer driver genes were found to show a mixed pattern with age in different cancer types. For instance, PTEN decreases with age in colon and endometrial, but increases in cervical cancer. CDH1 is more frequent in younger stomach cancer patients, but more common in older breast cancer patients. Could these be explained by the presence of different subtypes of cancers? The observed cancer-specific pattern seems to suggest that more complex subtyping is likely to be the cause. Additionally, are there any genes that showed consistent pattern across different cancer types? If so, these may have stronger support as age-specific mutations. For the disease with higher numbers of age-DEGs, are these more heterogeneous diseases? Whereas the 5 diseases with no significant gene could be more homogeneous.

2. One major finding is that cancer in female reproductive organs, including breast, ovarian and endometrial cancers, are among those with the highest number of age-DEGs and age-DMGs. -- Could this be caused by the presence of germline pre-disposition mutations, such as BRCA, in certain patients?

3. Mutational signature is certainly an important measurement of age-related mutations. It was not clear why this analysis was not included in the current study.

Minor:

1. For typical SCNA analysis in cancer, SCNAs were identified by comparing paired tumor/normal samples. Could some of the recurrent age-related focal SCNAs across different cancer types are indeed caused by age-related changes of the common germline sample (e.x. blood) in opposite direction? For example, an age-related copy number loss in blood would lead to all the matched tumor sample to show a gain in that region?

2. Was the quantitative difference of mutations taken into consideration? For example, one-copy loss vs homozygous loss, and mutations' clonal statuses or VAF. If these are over-complicated to be included in the model, that would need to be clarified.

3. Line 132. For the finding that lung adenocarcinoma was the only cancer with a negative association between overall SCNA and age, could this be caused by the difference in patient smoking

history?

4. Line 140: in cervical cancer...a significant association between age and chromosome/arm-level. However, in figure 2b, CESC does not seem to be significant?

5. In endometrial cancer, if we exclude all tumors with MSI-H, or mutations in POLE or POLD1, do we then see a positive correlation between age and mutation burden?

6. Line 146, a potential typo of "Fig. Fig. 2d, e"

Reviewer #3 (Remarks to the Author): Expert in bioinformatics, transcriptomics, and epigenetics

In this manuscript, the users investigated age-related differences in genomic instability, somatic copy number alterations (SCNAs), somatic mutations, pathway alterations, gene expression, and DNA methylation landscape across various cancer types. They showed that, in general, genomic instability and mutations frequency increase with age, and identified several age-associated genomic alterations in cancers, particularly in low-grade glioma and endometrial carcinoma. They also demonstrated that age-related gene expression changes are controlled by age-related DNA methylation changes and that these changes are linked to numerous biological processes. In general, the paper is well written and the idea is clear. Below are comments and suggestions for analysis that to strengthen the paper.

- Can the authors give a potential explanation for the negative correlation between percent genomic LOH and age for lung adenocarcinoma, oesophageal, and liver cancer?
- Can the authors give a potential explanation for the negative association between the overall SCNA score and age for lung adenocarcinoma?
- Can the authors give a potential explanation for the increase of gain of specific chromosome arms with age in certain cancer types and the decrease of gain of other chromosome arms with age in other cancer types? For instance, do these chromosome arms contain certain genes that drive these cancer types? And similarly, can they comment on the increase and decrease of loss of chromosome arms with age for other cancer types.
- Can the authors provide more details about the functional roles of the genes that showed negative correlation between mutation and age in low grade glioma and glioblastoma? Do these genes share some functional roles together?
- Figure S5 shows that the relationship between age and somatic mutations overall is positive. There are 6 substitution classes. Does this positive relationship mostly contribute by C->T mutations as suggested by Line 196?
- What is the relationship between cancer-related changes (DEGs & DMCs) and the age-related changes (DEGs & DMCs)? And what is the interplay among somatic mutations, DMCs, and DEGs in

the aging context?

- In Figure 6e, several pathways that are not directly related to cancer was included such as exon development. How did the authors perform the GSEA analysis for methylation changes? ClusterProfiler was used as mentioned in the methods section without much details. Were DMC associated genes used for GSEA analysis? If so, how was the gene associated with DMCs? How did the authors choose the background for GSEA analysis? Not all genes were covered equally across the promoters, how did the author make sure the GSEA analysis was not influenced by the coverage of CpGs in the methylation profiling?

- Minor comments:

- o On page 9, line 222, the users used “UCEC” without mentioning what it stands for.

- o In all the figures, the font size of legends and axes is very small. It negatively impacts the readability.

- o Line 146: there is an extra “fig.”

- o Line 170: missing citation for “previously identified cancer driver genes”.

Authors' response to the Reviewers' comments

Manuscript: NCOMMS-20-34538

Title: An Integrative Analysis of the Age-Associated Genomic, Transcriptomic and Epigenetic Landscape across Cancers

Dear Editor,

Thank you for considering our manuscript for publication in *Nature Communications*. We carefully read all the comments and suggestions from the editors and reviewers and revised the manuscript accordingly. Overall, we agree with the majority of the comments and revised the manuscript following the suggestions. In the few cases we chose not to fully follow the suggestions, our reasons for doing so are carefully detailed.

What follows are our point-by-point responses (in blue and marked with ####) to the comments from the editors and reviewers. Changes in the manuscript are highlighted in yellow.

Reviewer #1 (Remarks to the Author): Expert in ageing and cancer

Chatsirisupachai and colleagues present integrated analyses of TCGA data to describe how genomic, epigenetic and gene expression changes in cancers are dependent on the age of the patient. Mostly, they performed linear regression analyses on various genetic, epigenetic and gene expression parameters across all cancers or individual cancers as a function of age. I believe that such a method has advantages over binning (e.g. segregating cases into halves, quartiles, etc), as one would expect progressive changes that would mirror other progressive changes associated with aging.

Genomic instability (GI) indices and LOH show age-dependent increases in multiple cancers, although it's interesting that some cancer types show the opposite trend (such as with LOH and lung AC). Similarly, increases in whole genome duplication (WGD) with age was observed across cancer types, with significant increases for some cancers like endometrial and ovarian. Notably, endometrial cancers show increases in GI, LOH and WGD with age. Consistent with the observations for GI, somatic copy number alterations (SCNA) also increased with age, evident in similar cancers as observed for GI (plus some additional ones). They also identified the chromosomal arms that tend to be gained and lost more often with age, as well as focal SCNA that were altered with age (again, endometrial cancers, ovarian cancers and low-grade gliomas stand out). Again, it is interesting that lung AC shows the opposite pattern, with decreased SCNA in cancers from older individuals. Cancers with SCNAs altered by age also exhibited corresponding alterations in the expression of known cancer driver genes.

Not surprisingly, all cancers combined and 18 individual cancers exhibit a significant increase in point mutational burden with age, likely due to the increased spontaneous deamination driven C>T changes that characterize aging tissues. The only exception was

endometrial cancers, and they showed that younger endometrial cancers are more frequently MSI-H, and have more frequent POLE and POLD1 mutations (hypermutators). They also identify particular genes where mutation frequency is impacted by age (excluding hypermutator cancers); for example, IDH1 mutations in gliomas negatively correlate with age. It's interesting that mutations in the same gene can negatively and positively correlate with age for different cancers. These analyses highlight how age-dependent changes in selective pressures could interface with mutational changes to impact the mutational landscape of cancers.

They then examined age-associated changes in oncogenic signaling pathways in cancers. 5 of 10 signaling pathways that were previously shown to be deregulated in cancers showed age dependence increases in these pathways across cancers (more frequent mutational alterations or SCNA in cancers with increasing age). They extended these analyses, showing pathway alterations across 15 cancer types. Thus, selection for mutational deregulation of different pathways appears to be differential across various cancers as a function of age.

For some cancers like gliomas, a large number of genes show differential expression dependent on age, while for other cancers there were no such genes. Clearly, some cancers like gliomas exhibit very different biology dependent on the age of the host, while others less so. Changes in gene expression were negatively concordant with those for DNA methylation, as would be expected, although the strong correlations observed suggest a major role for such DNA methylation in controlling age-dependent changes in gene expression in cancers. They observed similar negative correlations between gene expression and DNA methylation for some Gene Ontology terms.

In all, these bioinformatic analyses of TCGA data to determine the age-dependence of mutational, epigenetic and gene expression changes in cancers is thorough and valuable. The assumptions made and other considerations (such as the use of cutoffs in terms of case #s for some analyses) appear well-justified, and their interpretations of the data appear reasonable. The Intro and Discussion both define the problems and offer a good analysis of the implications of their results (such as to why female reproductive cancers show such clear age-dependence in epigenetic parameters). The Methods are clear and seem appropriate (although genomic bioinformatics is not my area of expertise). Obviously, a comprehensive analysis of genetic and expression changes dependent on age across cancers is going to raise more questions than it answers, providing fodder for hypothesis testing in the years to come. And that's what makes this study very useful.

These analyses will serve as a valuable resource for investigators interested in associations between age and cancers. They also highlight the important role that age plays in sculpting the mutational and phenotypic landscapes of cancers, influenced both by selection and mutational processes. This study clearly shows that the age of the patient matters when it comes to the biology of the cancer, which should have profound implications for how we develop preventative, diagnostic and therapeutic interventions.

Finally, as they cite, a study deposited on bioRxiv by Li et al used TCGA and PCAWG databases to perform a similar analysis for how aging impacts cancer genomics. The current

study and this study find some common patterns, but due to their different methods also make unique discoveries (for example, related to the global transcriptomic and DNA methylation changes revealed from the current study). In the end, the field will be well served by the availability of both studies, leading to stronger conclusions overall and highlighting some unique contributions from each study.

We are thankful for the positive comments from the reviewer. We are pleased in particular that the reviewer finds our study a valuable resource for scientists interested in ageing and cancer.

Minor:

1) Suppl Fig 5A, showing the increased point mutational burden with age of cancer (all combined), should be included in a main figure (like Fig 4). I imagine that the authors chose to put it supplemental given that this association is known, but if it can be squeezed into to a main figure, it will help the reader better appreciate the full picture of mutational changes with age.

Yes, we initially chose to include it in a supplementary figure because this is a well-known association. We agree with the reviewer that by including it in the main figure instead would provide the reader a better picture of age-associated mutational changes. The figure is now included in Fig 4a.

2) The GI score is very well described in the Methods, but it would be helpful if the authors very briefly described the basics of this score in the Results section when these analyses are presented.

We thank the reviewer for pointing this out. We have now included a brief description in the Results section at Line 81 as follows.

We first derived genomic instability (GI) scores, calculated as the genome fraction (percent-based) that does not fit the ground state, defined as 2 for tumours that have not undergone whole-genome duplication (WGD), and 4 for tumours that have undergone WGD (Methods).

Reviewer #2 (Remarks to the Author): Expert in cancer genomics

In the current study, Chatsirisupachai et al. reported an integrative analysis of TCGA data to identify age-associated patterns of somatic alterations in human cancers. This is an important research question given that aging is the #1 risk factor of cancer. The analysis included multiple data types such as copy number, LOH, somatic mutations, expression, and methylation. Overall, this is a very large multi-omics study that provides novel insights into cancer and aging. The findings are scientifically interesting and clinically relevant. However, there are also some technical questions, many of which are related to the complexity of the disease, that could be better addressed.

We are happy that the reviewer finds our data are scientifically interesting and clinically relevant. We would also like to thank the reviewer for the thoughtful comments and we agree that these issues are important. We provide the point-by-point response below.

Major:

1. For the observed age-related genomic differences that are specific to certain cancers, for instances, the more frequent hypermutators in younger endometrial cancer patients, and IDH-WT vs IDH-mutant in glioma, do they represent unique features of aging, or age-related subtypes of certain cancer? Either could be important, but may need to be distinguished or discussed carefully. The latter could be explained by a heterogeneous nature of the disease itself that may or may not have been discovered. Similarly, for somatic mutations, certain cancer driver genes were found to show a mixed pattern with age in different cancer types. For instance, PTEN decreases with age in colon and endometrial, but increases in cervical cancer. CDH1 is more frequent in younger stomach cancer patients, but more common in older breast cancer patients. Could these be explained by the presence of different subtypes of cancers? The observed cancer-specific pattern seems to suggest that more complex subtyping is likely to be the cause. Additionally, are there any genes that showed consistent pattern across different cancer types? If so, these may have stronger support as age-specific mutations. For the disease with higher numbers of age-DEGs, are these more heterogeneous diseases? Whereas the 5 diseases with no significant gene could be more homogeneous.

We would like to thank the reviewer for this comment. We agree with the reviewer that distinguishing between ageing features and age-related subtypes is important. However, cancer subtypes can be difficult to define and can be subjective. There are numerous ways to define subtypes, for example, by some markers such as ER+ and ER- breast cancer, by histological subtypes or by gene expression or methylation unsupervised clustering.

Here, we tried to address this issue by considering some standard and broadly recognised subtypes, but we would also like to note that there might be several other ways to define subtypes within each cancer type.

Firstly, the strong age-related features we found in IDH-WT/IDH-mutant represent previously identified subtypes in gliomas. The IDH-mutant subtype is predominantly found in younger patients¹⁻³. Thus, this feature could represent age-associated subtypes. Similarly,

endometrial cancer consists of four subtypes previously identified, including *POLE* (ultramutated), MSI-H (hypermuted), copy-number low (endometrioid) and copy-number high (serous-like)⁴. The *POLE* and MSI-H subtypes are dominated by mutational signatures associated with *POLE* mutation and MSI-related mutational processes. The other subtypes, however, displayed dominant COSMIC mutational signature 1 associated with ageing⁵. Therefore, the more frequent hypermutators in younger endometrial tumours are likely to represent age-associated subtypes.

Regarding the somatic mutations that showed an opposite pattern with age between cancer types, we checked whether mutations in some of these genes exhibit a feature of well-known subtypes in cancers or not. One example, as mentioned by the reviewer is the *CDHI* mutation which was enriched in the genomically stable (GS) stomach tumours, which were diagnosed at an earlier age⁶. Indeed, by investigating 306 stomach tumours with subtype information from a previous TCGA study⁷, we also found the significantly younger age at diagnosis in GS subtype comparing with other subtypes (Wilcoxon test, p-value = 0.0058, **Fig. 1 Top left**), and an enrichment of tumours with *CDHI* mutations in this subtype (Fisher's exact test, p-value = 2.0×10^{-5} , **Fig. 1 Top right**). In breast cancer, although we have already included oestrogen receptor status, ER+ or ER- tumours, in our linear model, other cancer subtyping systems might affect the age-associated features. A previous study has reported enrichment of *CDHI* loss-of-function mutations in invasive lobular carcinoma (ILC), a second most common histological subtype⁸. Here, using the subtype information from a previous study⁹, we investigated 734 breast cancer samples that have been classified as ILC, invasive ductal carcinoma (IDC), mixed and other histological subtypes. We showed that ILC was diagnosed at an earlier age comparing with all other subtypes (Wilcoxon test, p-value = 0.00081, **Fig. 1 Middle left**), as well as an enrichment of *CDHI* mutations in ILC (Fisher's exact test, p-value = 4.4×10^{-38} , **Fig. 1 Middle right**). Apart from *CDHI*, we also explored the association between gene mutations and ILC subtypes in other cancer driver genes that we found age-associated patterns in breast cancer, including *PIK3CA*, *GATA3*, *KMT2C* and *MAP3KI*. We also found that *PIK3CA* mutations were more frequent in ILC (Fisher's exact test, p-value = 0.0013, **Fig. 1 Bottom left**), while *GATA3* mutations tended toward lower frequency in ILC (Fisher's exact test, p-value = 0.054, **Fig. 1 Bottom right**) as suggested by a previous study⁸. However, there was no significant association between ILC subtype and mutations in *KMT2C* and *MAP3KI* (Fisher's exact test, p-value = 0.16 in both genes).

Another example the reviewer mentioned was an increased *PTEN* mutation with age in cervical cancer, we found no difference in the proportion of *PTEN* mutations between two cervical cancer subtype, adenocarcinoma and squamous cell carcinoma (Fisher's exact test, p-value = 1, **Fig. 2 Right**). Age at diagnosis was not different between both subtypes (**Fig. 2 Left**). This could either suggest an age-associated feature regardless of subtype or an unidentified subtype that needs to be further investigated. For the decrease in *PTEN* mutations with age in colon cancer and endometrial cancer, this might be due to the presence of MSI-H tumours in the data. Although we have already excluded tumours with more than 1,000 mutations, some MSI-H tumours contain less than 1,000 mutations and still present in the data. To better address this bias, we re-analysed our age-associated mutations by excluding all tumours harbouring more than 1,000 mutations per exome and MSI-H tumours.

After we removed MSI-H tumours from the analysis, *PTEN* mutation was not significantly associated with age in both colon and endometrial cancer.

Fig.1 Top left: Genomically stable (GS) stomach cancers (STAD) are presented in younger patients. **Top right:** An enrichment of *CDH1* mutations in GS subtype. **Middle left:** Invasive lobular carcinoma (ILC) breast cancers (BRCA) are presented in older patients. **Middle right:** An enrichment of *CDH1* mutations in ILC subtype. **Bottom left:** An enrichment of *PIK3CA* mutations in ILC subtype. **Bottom right:** Mutations in *GATA3* are less frequent in ILC subtype.

Fig. 2 Left: Age at diagnosis of cervical cancer (CESC) subtypes, adenocarcinoma and squamous cell carcinoma, are not different. **Right:** The frequency of PTEN mutations in cervical adenocarcinoma and squamous cell carcinoma are not different.

In the somatic mutation analysis, we found five genes showing a consistent pattern with age. Mutations in these genes, however, associated significantly with age only in two cancer types, including lower mutations of *ATRX* and *IDH1* in low-grade gliomas and glioblastoma with age, an increased mutation of *NFI* with age in Pheochromocytoma and Paraganglioma and melanoma an increased mutation of *NOTCH1* in cervical cancer and head and neck cancer, and an increased mutation of *PIK3CA* in breast cancer and cervical cancer. From this information, there is no universal gene that shows an increased or decreased mutation with age across several cancer types. One major reason is that the prevalence of a cancer driver gene mutated in different cancer types is vastly different. Additionally, it is still unclear whether different tissues age similarly or differently. Therefore, the increase or decrease in the frequency of mutations with age for cancer from each tissue could be due to differences in tissue environments and the interplay between cells harbouring mutations and tissue environment changes with age that needs to be further investigated both computationally and experimentally.

As the reviewer pointed out, we observed a vast difference between the number of age-DEGs from different cancer types. However, it is difficult to conclude that this difference is caused by the homogeneity or heterogeneity of diseases. In fact, pointing out that one cancer is more homogeneous or more heterogeneous than other cancers can be tricky. For example, by histological type, most cases in the TCGA ovarian cancer cohort are ovarian serous cystadenocarcinoma. It is, however, among one of the top cancers with the highest number of age-DEGs. On the other hand, there were no age-DEGs in stomach cancer, which consists of different histological subtypes such as mucinous type, diffuse type and papillary type or consists of four molecular subtypes distinguished by genomic data, including EBV+, MSI,

GS, and chromosomal instability (CIN). Therefore, the numbers of age-DEGs could be independent of the cancer heterogeneity.

Overall, we agree with the reviewer that it is interesting and important, yet difficult, to untangle the age-associated features within a subtype from the age-associated subtypes. Here, while we were able to explain some of the age-associated mutations in relation to age-related subtypes, further detailed investigation in each cancer type is required to fully distinguish age-related subtype effects from age-associated effects, or even to identify a new age-related subtype. Our perspective, however, is that either age-associated mutations come from age-related subtypes or from age-related features could reflect the biological differences between cancers at different ages. Thus, our current manuscript underscores the importance of age in cancer research and calls for more studies in relation to ageing and cancer. We incorporated some of these examples we addressed above in the Results section of the manuscript at Line 302.

Mutations in CDH1 were more frequent in younger stomach cancer patients (OR = 0.9414, 95%CI = 0.9027-0.9800, adj. p-value = 0.006), but more common in older breast cancer patients (OR = 1.0218, 95%CI = 1.0049-1.0392, adj. p-value = 0.0183) (Fig. 4f). This result highlights cancer-specific patterns of genomic alterations with age. We tested whether age-associated subtypes could explain differences in mutation with age. Using subtype information from a previous TCGA study⁴⁰, CDH1 mutations were found more often in the genomically stable (GS) subtype of stomach cancer (two-sided Fisher's exact, p-value = 2.0×10^{-5}), which was presented more frequently in younger patients (two-sided Wilcoxon rank sum test, p-value = 0.0058) (Supplementary Fig. 10a-b). As expected, CDH1 mutations were highly enriched in the invasive lobular carcinoma (ILC) subtype of breast cancer³⁵ (two-sided Fisher's exact, p-value = 4.4×10^{-38}), which was more prevalent in older patients (two-sided Wilcoxon rank sum test, p-value = 0.00081) (Supplementary Fig. 10c-d).

We also discussed this point in the Discussion section at Line 472 as follows.

Indeed, some of these age-related differences might be explained by age-related subtypes, such as the high prevalence of CDH1 mutation in invasive lobular breast carcinoma and genomically stable stomach cancer, that are presented more often in older and younger patients, respectively. Further detailed investigation in each cancer type is required to fully distinguish age-related subtype effects from age-associated effects, or even to identify a new age-related subtype. Our perspective is that whether age-associated genomic differences come from age-related subtypes or from age-related features they reflect biological differences between cancers at different ages, highlighting the impact of age on the molecular profile of cancer.

2. One major finding is that cancer in female reproductive organs, including breast, ovarian and endometrial cancers, are among those with the highest number of age-DEGs and age-

DMGs. -- Could this be caused by the presence of germline predisposition mutations, such as BRCA, in certain patients?

We thank the reviewer for raising this point. To address this, we retrieved the information of germline predisposition in TCGA patients from the previous TCGA study¹⁰. We then excluded samples harbouring germline variants located in exons of *BRCA1*, *BRCA2* and *TP53* that did not overlap with variants reported in the 1000 Genomes project. In total, the number of samples excluded from each TCGA cohort in gene expression are 134 out of 1,011 (13.25%), 63 out of 288 (21.88%) and 56 out of 432 (12.96%) for breast (BRCA), ovarian (OV) and endometrial cancers (UCEC), respectively. Similarly, the number of samples excluded from each TCGA cohort in methylation are 93 out of 719 (12.93%), 101 out of 545 (18.53%) and 47 out of 360 (13.06%) for BRCA, OV and UCEC, respectively. We next repeated our multiple linear regression analyses to identify age-DEGs and age-DMGs using samples without germline variants. For gene expression analysis, significant age-DEGs (adj. p-value < 0.05) identified from samples without germline variants largely overlapped with age-DEGs from all samples (**Fig. 3**). Indeed, we also observed a high correlation between regression coefficients of multiple regression analysis from all tumours and tumours without samples harbouring germline predisposition variants in BRCA (R = 0.98, Pearson correlation), OV (R = 0.93) and UCEC (R = 0.98) (**Fig. 4**).

Likewise, we found a large overlap between age-DMGs identified from all samples and samples without germline variants in all three cancers (**Fig. 5**). A high correlation between regression coefficients of multiple regression analysis from all tumours and tumours without samples containing germline variants were observed in BRCA (R = 0.99, Pearson correlation), OV (R = 0.97) and UCEC (R = 0.98) (**Fig. 6**).

Due to the similarity between age-DEGs and age-DMGs identified from all patients and only patients without germline predisposition mutations, the presence of germline mutations is unlikely to be the main cause of the high number of age-DEGs and age-DMGs. As we discussed at Line 500, other factors such as hormonal changes could be responsible for the age-associated gene expression and methylation changes. We have incorporated this data into the Results section of the revised manuscript at Line 370 and Supplementary Figure 12 as follows.

To exclude the possibility that germline predisposition mutations in some patients may cause such a high number of age-DEGs and age-DMGs in cancers of the female reproductive system, we excluded samples harbouring germline mutations in *BRCA1*, *BRCA2* and *TP53* as previously identified⁴² from the breast, ovarian and endometrial cancer cohorts and repeated the multiple linear regression analysis. We observed a high correlation between regression coefficients of the analyses from all tumours and the analyses excluding samples with germline mutations, for all three cancer types and for both gene expression and methylation (R = 0.93-0.99, p-value < 2.2×10^{-16}) (Supplementary Fig. 12). The overlap between age-DEGs or age-DMGs identified from all samples and from samples without germline variants were large (Supplementary Fig. 12, Supplementary Data 15). Therefore, the high number of age-DEGs and age-DMGs are independent of the presence of germline predisposition mutations in some patients. We, therefore, used age-DEGs and age-DMGs identified from all samples for subsequent analyses.

Fig. 3 Overlap between age-DEGs identified from all samples (adj. p-value < 0.05) and samples without germline variants (adj. p-value < 0.05).

Fig. 4 Pearson correlation between regression coefficient from the multiple linear regression analysis to identify the association between gene expression and age in all samples and in samples without germline variants. Each dot represents one gene.

Fig. 5 Overlap between age-DMGs identified from all samples (adj. p-value < 0.05) and samples without germline variants (adj. p-value < 0.05).

Fig. 6 Pearson correlation between regression coefficient from the multiple linear regression analysis to identify the association between methylation and age in all samples and in samples without germline variants. Each dot represents each gene.

3. Mutational signature is certainly an important measurement of age-related mutations. It was not clear why this analysis was not included in the current study.

We thank the reviewer for the suggestion. We agree with the reviewer that mutational signature analysis is an important analysis related to age-related mutations. However, our focus in the study was to identify cancer driver genes in which their mutations showed an age-associated pattern. Moreover, several studies have already reported the age-related mutational signatures using TCGA and PCAWG data, pointing mainly toward COSMIC signature 1 and signature 5 (clock-like mutational signatures)^{11,12}. Importantly, we are aware that this analysis has been performed by Li et al.¹³ that is also submitted to this journal. Thus, we are not sure performing a similar analysis will provide novel insights. If the reviewer and/or Editor insists we perform this analysis we could do it, however.

Minor:

1. For typical SCNA analysis in cancer, SCNAs were identified by comparing paired tumor/normal samples. Could some of the recurrent age-related focal SCNAs across different cancer types are indeed caused by age-related changes of the common germline sample (e.x. blood) in opposite direction? For example, an age-related copy number loss in blood would lead to all the matched tumor sample to show a gain in that region?

We thank the reviewer for raising this point. We performed a careful analysis of samples with potential copy number changes in the matched germline sample ($n=9,678$). We detected segments of more than 10Mb that have B-allele frequency (BAF) < 0.49 or > 0.51 in germline samples using the ASPCF function implemented in ASCAT. This yielded 58 samples (0.6 %) with at least one clear BAF deviation, indicating either sub-clonal copy-number variations or copy-number alterations from contaminated tumour cells. We repeated age-related focal SCNA analysis excluding these 58 samples and found that almost all significant age-associated regions remain significant. Only in one gain region, chromosome 12q14.1 in low-grade glioma, where the result changes from significant (adj. p-value = 0.048) to non-significant (adj. p-value = 0.077) after excluding one sample with the gain in this region from the analysis. We inspected the excluded sample to check the region where BAF is deviated from 0.5 and found that this sample has been flagged due to a mixed pattern between 1+1 and 2+0 at chromosome 9q, suggesting that this CNV in this germline blood sample does not affect the region with age-associated SCNA in cancer (12q14.1). The change in p-value we observed thus more likely to come from a reduced statistical power. Therefore, our analysis of age-related SCNAs was not confounded by the presence of somatic mosaicism in normal samples.

2. Was the quantitative difference of mutations taken into consideration? For example, one-copy loss vs homozygous loss, and mutations' clonal statuses or VAF. If these are over-complicated to be included in the model, that would need to be clarified.

No, we did not take homozygous/heterozygous and clonal statuses into consideration. We acknowledge that it would be an interesting analysis but considering that it would be

over-complicated. We agreed with the reviewer that it is important to clarify and have included a paragraph stating this in the revised manuscript at Line 480 as follows.

One limitation, however, of our age-related genomic analyses is that we did not include in our model information on quantitative differences of genomic alterations, such as homozygous/heterozygous loss and clonal/subclonal mutations, to not over-complicate the analyses.

3. Line 132. For the finding that lung adenocarcinoma was the only cancer with a negative association between overall SCNA and age, could this be caused by the difference in patient smoking history?

Yes, smoking is enriched in younger lung adenocarcinoma patients as shown in **Fig. 7** below. Patients that are current smokers are significantly younger than other groups. Although we included smoking status in our linear model, it is likely that smoking status still confounds the data. When we performed multiple linear regression on samples from lifelong non-smokers only, there is no association between age and overall SCNA score (p-value = 0.75) (**Fig. 8 Left** below). We, however, found a significant association between age and overall SCNA score in both current smokers and current reformed smokers (p-value = 0.0420 and 0.0421, respectively) (**Fig. 8 Right and Bottom** below). Therefore, although the presence of current smokers in younger patients could partly cause the negative association between overall SCNA score and age, other unknown factors might also contribute to this negative correlation.

In fact, we also found a negative association between age and mutational burden in lung adenocarcinoma using simple linear regression analysis. However, this negative association is no longer significant after we adjusted for smoking status. It is worth noting that the Li *et al.* study also reported negative associations between age and SCNA as well as mutational burden in lung adenocarcinoma¹³. We have now included this potential explanation of a negative association between SCNA score and age caused by the difference in patient smoking history in our manuscript at Line139 as follows.

lung adenocarcinoma is the only cancer type exhibiting a negative association between overall SCNA score and age (Fig. 2a, Supplementary Fig. 3a, and Supplementary Data 5), possibly due to the presence of current smokers in younger lung adenocarcinoma patients (Supplementary Fig. 2a). When we analysed only non-smokers, there was no significant association between age and overall SCNA score (Supplementary Fig. 4a). However, the significant negative association was found when we analysed only current reformed smokers and only current smokers (Supplementary Fig. 4b-c), thus other unexplained factors apart from smoking status might also contribute to this higher SCNA score in younger smokers.

Fig. 7 An enrichment of current smokers in younger lung adenocarcinoma (LUAD) patients. The group comparison was performed by the Kruskal-Wallis test. The pairwise comparisons were done using Dunn's test. P-values from Dunn's test between each pair of comparisons are shown.

Fig. 8 Left: Association between age and overall SCNA score in lifelong non-smoker lung adenocarcinoma patients. **Right:** Association between age and overall SCNA score in current reformed smokers. **Bottom:** Association between age and overall SCNA score in current smokers.

4. Line 140: in cervical cancer...a significant association between age and chromosome/arm-level. However, in figure 2b, CESC does not seem to be significant?

Thank you for pointing this out, it should be sarcoma, not cervical cancer. This has now been corrected.

5. In endometrial cancer, if we exclude all tumors with MSI-H, or mutations in POLE or POLD1, do we then see a positive correlation between age and mutation burden?

We investigated the association between age and mutational burden in endometrial cancer excluding all MSI-H and POLE/POLD1 tumours and found a significant increase in mutational burden with age (**Fig. 9** below). This result is included in the Supplementary Fig. 7c and Line 249 as follows.

Indeed, when we excluded tumours with MSI-H and tumours containing *POLE/POLD1* mutations from the analysis, we found a significant positive association between mutation burden and age in endometrial cancer (adj. R-squared = 0.12, p-value = 0.00138) (Supplementary Fig. 7c).

Fig. 9 Association between age and mutational burden in endometrial cancer excluding tumours with MSI-H, POLE or POLD1 mutations.

6. Line 146, a potential typo of “Fig. Fig. 2d, e”

Thank you for spotting this. It is corrected now.

Reviewer #3 (Remarks to the Author): Expert in bioinformatics, transcriptomics, and epigenetics

In this manuscript, the users investigated age-related differences in genomic instability, somatic copy number alterations (SCNAs), somatic mutations, pathway alterations, gene expression, and DNA methylation landscape across various cancer types. They showed that, in general, genomic instability and mutations frequency increase with age, and identified several age-associated genomic alterations in cancers, particularly in low-grade glioma and endometrial carcinoma. They also demonstrated that age-related gene expression changes are controlled by age-related DNA methylation changes and that these changes are linked to numerous biological processes. In general, the paper is well written and the idea is clear. Below are comments and suggestions for analysis that to strengthen the paper.

We are grateful that the reviewer finds our paper is well written, and the idea is clear. We would also like to thank the reviewer for thoughtful comments, which helped strengthen the manuscript. We here provide the point-by-point response below.

- Can the authors give a potential explanation for the negative correlation between percent genomic LOH and age for lung adenocarcinoma, oesophageal, and liver cancer?

We looked into the potential confounding factors in our linear model. Apart from age, tobacco smoking had a significant association with percent genomic LOH in lung adenocarcinoma. As shown in **Fig. 10** below, patients that were current smokers were younger than the other groups. Although we included smoking status in our linear model, it is likely that smoking status still confounds the data. The percent genomic LOH in tumours from current smokers is significantly higher than that of current reformed smokers, whereas there was no difference between percent genomic LOH in tumours from lifelong non-smokers and the other groups (**Fig. 11** below). While we found no association between age and LOH in lifelong non-smokers and current reformed smokers (**Fig. 12 Left-Right** below), the negative association was shown in current smokers (**Fig. 12 Bottom** below). Thus, it is possible that the negative correlation between percent genomic LOH and age in lung adenocarcinoma can partly be explained by the younger age of current smokers. However, the negative association between age and percent genomic LOH that is presented when analysing only current smokers indicates that other factors might also contribute to this negative correlation.

Fig. 10 An enrichment of current smokers in younger lung adenocarcinoma (LUAD) patients. The group comparison was performed by the Kruskal-Wallis test. The pairwise comparisons were done using Dunn’s test. P-values from Dunn’s test between each pair of comparisons are shown.

Fig. 11 Tumours from current smokers harbour a higher percent genomic LOH than that of current reformed smokers. The group comparison was performed by the Kruskal-Wallis test. The pairwise comparisons were done using Dunn’s test, multiple hypothesis correction was performed by Bonferroni method. Adjusted P-values from Dunn’s test between each pair of comparisons are shown.

Fig. 12 Left: Association between age and percent genomic LOH in lifelong non-smoker lung adenocarcinoma patients. **Right:** Association between age and percent genomic LOH in current reformed smokers. **Bottom:** Association between age and percent genomic LOH in current smokers.

For liver cancer, there was also a significant association between LOH and histologic grade, in addition to age. Grade 3 and 4 tumours were enriched in younger patients (**Fig. 13 Left** below). The percent genomic LOH in grade 3 and 4 tumours is significantly higher than that of grade 1 and 2 tumours (**Fig. 13 Right** below). When we investigated the relationship between age and percent genomic LOH in G1 and G2 tumours using multiple linear regression, no association was found ($p\text{-value} = 0.125$) (**Fig. 14 Left** below). However, the significant negative association between age and percent genomic LOH was shown in G3 and G4 tumours ($p\text{-value} = 0.0324$) (**Fig. 14 Right** below). Thus, although the negative correlation between age and percent genomic LOH in liver cancer can partly be explained by the higher proportion of G3 and G4 tumours in younger patients, there might be undiscovered factors related to age and LOH in liver cancer.

Fig. 13 Left: Tumours with histological grade G3 and G4 are presented in younger liver cancer (LIHC) patients. **Right:** G3 and G4 tumours harbour a higher percent genomic LOH. The group comparison was performed by the Kruskal-Wallis test. The pairwise comparisons were done using Dunn’s test, multiple hypothesis correction was performed by Bonferroni method. Adjusted P-values from Dunn’s test between each pair of comparisons are shown.

Fig. 14 Left: Association between age and percent genomic LOH in histological grade G1 and G2 liver cancers. **Right:** Association between age and percent genomic LOH in histological grade G3 and G4 liver cancers.

Next, our linear model for investigating the association between age and LOH in oesophageal cancer did not include race, pathologic stage and smoking history due to the missing data in more than 10% of samples (10.8%, 13.1% and 10.2% respectively). We then performed multiple linear regression adjusting for all variables used previously (gender, histologic grade and alcohol history) together with race, pathologic stage and smoking history. We found that the negative association between age and percent genomic LOH in oesophageal cancer is no longer significant (p-value = 0.197).

We briefly included the potential explanation for the negative association between percent genomic LOH and age for lung adenocarcinoma, oesophageal and liver cancer in the revised manuscript at Line108 as follows.

This negative correlation might be due to the difference in the distribution of age of samples with smoking status (lung adenocarcinoma and oesophageal cancer), race (oesophageal cancer) and tumour grade (liver cancer) (Supplementary Fig. 2), yet other unexplained factors might also contribute to the higher LOH level in younger patients in these three cancer types.

- Can the authors give a potential explanation for the negative association between the overall SCNA score and age for lung adenocarcinoma?

The negative association between SCNA score and age in lung adenocarcinoma is likely due to the difference in smoking status. As shown in **Fig. 10** above, there is an enrichment of current smokers in younger patients. Although we included smoking status in our linear model, it is likely that smoking status still confounds the data. When we performed multiple linear regression only samples from lifelong non-smokers, there is no association between age and overall SCNA score (p-value = 0.75) (**Fig. 15 Left** below). We, however, found a significant association between age and overall SCNA score in both current smokers and current reformed smokers (p-value = 0.0420 and 0.0421, respectively) (**Fig. 15 Right and Bottom** below). Therefore, although the presence of current smokers in younger patients could partly cause the negative association between overall SCNA score and age, other unknown factors might also contribute to this negative correlation.

In fact, we also found a negative association between age and mutational burden in lung adenocarcinoma using simple linear regression analysis. However, this negative association is no longer significant after we adjusted for smoking status. It is worth noting that the Li *et al.* study also reported negative associations between age and SCNA as well as mutational burden in lung adenocarcinoma¹³. We have now included this potential explanation of a negative association between SCNA score and age caused by the difference in patient smoking history in our manuscript at Line 139 as follows.

lung adenocarcinoma is the only cancer type exhibiting a negative association between overall SCNA score and age (Fig. 2a, Supplementary Fig. 3a, and Supplementary Data 5), possibly due to the presence of current smokers in younger lung adenocarcinoma patients (Supplementary Fig. 2a). When we analysed only non-smokers, there was no significant association between age and overall SCNA score (Supplementary Fig. 4a). However, the significant negative association was found when we analysed only current reformed smokers and only current smokers (Supplementary Fig. 4b-c), thus other unexplained factors apart from smoking status might also contribute to this higher SCNA score in younger smokers.

Fig. 15 Left: Association between age and overall SCNA score in lifelong non-smoker lung adenocarcinoma patients. **Right:** Association between age and overall SCNA score in current reformed smokers. **Bottom:** Association between age and overall SCNA score in current smokers.

- Can the authors give a potential explanation for the increase of gain of specific chromosome arms with age in certain cancer types and the decrease of gain of other chromosome arms with age in other cancer types? For instance, do these chromosome arms contain certain genes that drive these cancer types? And similarly, can they comment on the increase and decrease of loss of chromosome arms with age for other cancer types.

We thank the reviewer for this question. First of all, this age-associated arm-level copy number alteration analysis was performed on recurrently gain and loss arms in each cancer type identified by GISTIC2.0, therefore, these chromosome arms potentially harbour cancer driver genes. Gains of chromosome arms containing oncogenes, or loss of chromosome arms containing tumour suppressor genes are common mechanisms driving cancer. From our analysis, we identified an increase in the gain of chromosome 7 containing *EGFR* locus in glioblastoma and low-grade glioma. Moreover, gliomas also showed an increased chromosome 10 loss containing *PTEN* allele. The gain of chromosome 7 and loss of chromosome 10 are common features of IDH-WT gliomas, which occur more frequently in older patients. A recent analysis reported that chromosome 7 gain and 10 loss commonly

occurred at tumour initiation in glioblastoma, highlighting the importance of these arm-level events in driving this cancer type¹⁴.

For endometrial cancer, we found a large increase in chromosome arm loss with age. This might also be related to the difference in age-related subtypes, hypermutated with low copy-number alterations in tumours from younger patients and low mutations with high copy-number alterations in tumours from older patients. An example of arm-level loss in this cancer type included an increased loss of chromosome 17, where *TP53* located in, with age, although we did not find age-associated focal-level loss in *TP53*. Similarly, we identified an increase in loss of chromosome 9 with age, containing a tumour suppressor gene *CDKN2A*. However, we did not identify an increase in focal-level loss of this gene with age.

Conversely, some cancer-driver gene gain or loss were significantly increased or decreased with age only in focal-level, but not in arm-level alterations. For example, a gain of *ERBB2* increased with age, while chromosome 17, where *ERBB2* located, lost more frequently with age.

Apart from gliomas and endometrial cancer, arm-level gain and loss in other cancer types are also related to some of the known cancer-driver genes. For instance, we found an increase in chromosome 13q loss with age in thyroid cancer. This chromosome arm contains a tumour suppressor *RBI* gene. The gain of chromosome 12, harbouring the oncogene *KRAS*, increased with age in ovarian cancer. Chromosome 6 loss decreased with age in ovarian cancer, *QKI* gene coded for an RNA-binding protein located in this chromosome. A previous study reported the role of *QKI* on splicing aberration in ovarian cancer¹⁵. Another gene located in chromosome 6q is a tumour suppressor gene *MAP3K4*. The important roles of *MAP3K4* on cancer growth and epithelial-to-mesenchymal transition have been shown in intrahepatic cholangiocarcinoma¹⁶. Further studies are needed to investigate the role of loss of *MAP3K4* in ovarian cancer.

Indeed, while we can explain some of the age-associated chromosomal arm alterations, further closer inspection on the arm-level alterations is required to fully explain why some specific arms are more frequent or less frequent gains or losses in particular cancer types. We have incorporated some of these explanations in the Results section at Line 166 as follows.

These arms included 9p and 17p, containing tumour suppressor genes *CDKN2A* and *TP53*, respectively. Low-grade glioma and ovarian cancer, two other cancer types for which we found the highest significant association between age and SCNA scores, also exhibited a significant increase or decrease in losses with age in multiple arms (Fig. 2e-f, Supplementary Fig. 5). We also observed that losses of chromosome 10p and 10q increased with age in gliomas. Recurrent losses of chromosome 10 (containing *PTEN*), together with gains of chromosome 7 (containing *EGFR*) are important features in IDH-wild-type (IDH-WT) gliomas²⁴. This type of glioma was more common in older patients, whereas IDH-mutant gliomas were predominantly found in younger patients. Apart from gliomas and endometrial cancer, arm-level gains and losses in other cancer types are also related to known cancer-driver genes. For instance, we found an increased incidence in the loss of chromosome 13q (harbouring *RBI*) with age in thyroid cancer. Gains of chromosome 12 (containing the *KRAS* oncogene), increased with age in ovarian cancer. Indeed, while we can explain some of the

age-associated chromosomal arm alterations, further closer inspection of arm-level alterations is required to fully explain why some specific arms are more or less frequently gained or lost as a function of age in particular cancer types.

- Can the authors provide more details about the functional roles of the genes that showed negative correlation between mutation and age in low grade glioma and glioblastoma? Do these genes share some functional roles together?

Three genes that showed a negative correlation between mutation and age in low-grade glioma and glioblastoma are *IDH1*, *TP53* and *ATRX*. A recent functional study in neural stem cells (NSCs) by introducing *IDH1* mutation, *TP53* shRNA and *ATRX* shRNA showed that the combination of these three oncogenic hits blocks NSC differentiation and promotes gliomagenesis¹⁷. The transcriptional profile of their 3-hit NSC model resembled *IDH1*-mutated human low-grade astrocytomas, a subtype of low-grade gliomas. This 3-hit NSCs caused hypermethylation of CTCF motifs flanking the *SOX2* locus, disrupted chromatin looping, leading to the transcriptional downregulation of *SOX2*, a transcription factor important in self-renewal and differentiation of NSC, and therefore impaired NSC differentiation. In addition, this study showed that *IDH1* mutation alone is enough to cause a hypermethylated epigenetic state. However, slow cell growth and increased cell death were observed in *IDH1*-only mutant cells. P53 knockdown is important to overcome growth arrest and cell death. Another recent report demonstrated that *ATRX* inactivation caused epigenomic remodelling and shifting differentiation state¹⁸. The role of *ATRX* deficiency in alternative lengthening of telomeres, a telomerase-independent mechanism, has also been observed¹⁹. Taken together, three genes with a negative correlation between mutation and age in gliomas, *IDH1*, *TP53* and *ATRX* cooperatively promote gliomas by causing epigenomic alterations, impaired differentiation and evading growth arrest and cell death. We have incorporated this information in the Discussion section at Line 449 as follows.

A recent functional study in neural stem cells (NSCs) showed that the combination of *IDH1*, *ATRX* and *TP53* blocks NSC differentiation by causing hypermethylation of CTCF motifs flanking the *SOX2* locus, disrupting chromatin looping and dysregulation of *SOX2*, an important transcription factor in self-renewal and differentiation of NSCs⁵². Impaired differentiation, growth arrest evasion by mutations in *TP53*, and alternative lengthening of telomeres by *ATRX* inactivation thus cooperatively promote gliomas in younger patients.

- Figure S5 shows that the relationship between age and somatic mutations overall is positive. There are 6 substitution classes. Does this positive relationship mostly contribute by C->T mutations as suggested by Line 196?

Yes, this positive relationship was mostly contributed by C>T mutations. We performed multiple linear regression to investigate the association between age and the fraction of contribution (percentage of each substitution class in total base substitution) adjusting for gender, race and cancer type. The positive association is found only in C>T mutations

(regression coefficient = 0.058, p-value = 8.57×10^{-7}). We also found a negative association in C>A mutations (regression coefficient = -0.065, p-value = 8.84×10^{-10}). Fraction of contribution in the other classes was not significantly associated with age (p-value > 0.05). It should be noted that the positive and negative association we found between age and C>T and C>A substitutions, respectively, have previously been reported by Milholland et al²⁰. We also performed linear regression to assess the association between age and fraction of contribution of each substitution class in each cancer type. Consistent with the pan-cancer analysis, C>T mutations showed a significant positive association with age in six cancer types, whereas C>A mutations had a significant negative association with age in three cancer types (Fig. 14). This result was added into the Results section at Line 223 of a revised manuscript as follows.

This increase in mutation load was mainly contributed by C>T mutations, as we found a positive association between the fraction of C>T mutations and age (regression coefficient = 0.058, p-value = 8.57×10^{-7}) in a pan-cancer analysis. Conversely, the fraction of C>A mutations was negatively associated with age (regression coefficient = -0.065, p-value = 8.84×10^{-10}) (Supplementary Data 10), concordant with a previous report⁴. We also examined, for each cancer type, the association between age and fraction contribution of each substitution class. Consistent with the pan-cancer analysis, C>T mutations showed a significant positive association with age in six cancer types, whereas C>A mutations had a significant negative association with age in three cancer types. (Supplementary Fig. 7b, Supplementary Data 10).

Fig. 14 Cancer type-specific association between age and fraction contribution of substitution class.

- What is the relationship between cancer-related changes (DEGs & DMCs) and the age-related changes (DEGs & DMCs)? And what is the interplay among somatic mutations, DMCs, and DEGs in the aging context?

We previously²¹ explored the relationship between differentially expressed genes with ageing and differentially expressed genes in cancer (cancer vs. normal) in nine tissues. In general, the gene expression in cancer and ageing change in opposite direction in most tissues, except in thyroid and uterus²¹. The overall opposite gene expression patterns between age-related and cancer-related changes also found by another study²². The relationship between DNA methylation changes in ageing and cancer is not straightforward. In general, global hypomethylation occurs in both ageing and cancer, while hypermethylation happens in promoter regions of specific genes²³. It has been proposed that methylation changes occur in a subpopulation of cells during normal ageing, might predisposing them to be more susceptible to tumour-initiating mutations²⁴. A recent study, however, reported differences in DNA methylation changes in ageing and cancer. While hypermethylated regions in both ageing and cancer are associated with bivalent chromatin signatures, hypomethylated sequences in ageing and cancer were enriched at regions with active H3K4me1 mark and repressive H3K9me3 mark, respectively²⁵. Another paper reported only a small overlap between differentially methylated regions in ageing and cancer²⁶. Thus, the relationship between cancer-related methylation and age-related methylation changes is still unclear.

Somatic mutations accumulate during ageing, and clonal expansions occur, as recently shown in many tissues²⁷⁻²⁹. Mutations that provide a selective advantage for a cell over surrounding cells are selected, similar to clonal expansion in cancer. Depending on the gene in which mutations accumulated, gene expression and methylation level change as a result of somatic mutations. In addition, mutations that occur on a DNA sequence in gene regulatory networks could cause an aberrant gene expression, leading to an increase in transcriptional noise as well as defects in cell signalling with age³⁰. Indeed, even though it is beyond the scope of the current work, the interplay between somatic mutations, DMGs and DEGs in ageing is an important question waiting for more research. Unlike cancer, a study of somatic mutations in ageing has previously been limited by the challenge in detection, because mutations are recurrently present only in a small number of cells³⁰. A recent advance in ultra-deep sequencing and *in vitro* clonal expansion followed by sequencing opens a great opportunity to study somatic mutations in ageing^{28,31}, and their influence on gene expression and epigenetics in the near future.

- In Figure 6e, several pathways that are not directly related to cancer was included such as exon development. How did the authors perform the GSEA analysis for methylation changes? ClusterProfiler was used as mentioned in the methods section without much details. Were DMC associated genes used for GSEA analysis? If so, how was the gene associated with DMCs? How did the authors choose the background for GSEA analysis? Not all genes were covered equally across the promoters, how did the author make sure the GSEA analysis was not influenced by the coverage of CpGs in the methylation profiling?

We thank the reviewer for pointing out the intricacies of GSEA analysis, and allowing us to clarify. First of all, in methylation analysis, we used the one-to-one mapping genes and probes by selecting the probes that are most negatively correlated with the corresponding gene expression. By this procedure, we can perform GSEA as in gene expression. GSEA was done by using all genes in the analysis, ranking from the gene with the most positive regression coefficient with age (most up-regulated methylation in older patients) to the gene with the most negative regression coefficient with age (most up-regulated methylation in younger patients). Unlike over-representation analysis such as GO enrichment analysis that is based on a predefined gene set (for example, up-regulated genes in older patients) and hypergeometric test to identify the significant overlap between a predefined gene set and a gene set of interest, a ranked gene list consisting of all genes is used in the GSEA and the test was to determine whether member genes in a gene set of interest are randomly distributed throughout the ranked gene list or found more often in the beginning (higher in older patients in our case) or the bottom of the list (higher in younger patients in our case)³², thus, we did not have to choose a background for GSEA. We acknowledge the reviewer's concern that not all genes were covered equally across the promoter, and we cannot exclude the possibility that GSEA analysis using methylation data could be influenced by the probes that we did not include in our analysis. However, these enriched pathways in our results are mainly pathways that were enriched when we analysed gene expression data. And gene expression is a final readout from epigenetic control by DNA methylation. Despite the potential noise in GSEA analysis on methylation data, we, therefore, intended to include this result to emphasise the general opposite trend between gene expression and methylation when genes are ranked based on the fold change with age, suggesting potential age-associated epigenetic control of age-related gene expression in cancers.

We briefly mentioned this limitation in the Discussion section at Line 506 as follows.

A limitation of this analysis is that we chose only one methylation probe per gene to create a one-to-one mapping between genes and probes. Other probes might also have an impact on gene expression as well as might cause noise in our GSEA analysis from methylation data.

We also added more detail on the GSEA analysis in the Methods section at Line 778 as follows.

Gene Set Enrichment Analysis (GSEA) was performed to investigate the Gene Ontology (GO) terms that are enriched in tumours from younger or older patients. The analysis was done using the package *ClusterProfiler* (version 3.14.3)⁷⁸. Briefly, genes or methylation probes were ranked based on their regression coefficient with age from the most positive regression coefficient with age (most up-regulated in older patients) to the gene with the most negative regression coefficient with age (most up-regulated in younger patients). A ranked gene list consisting of all genes was used in the GSEA to determine whether member genes in a gene set of interest from GO are randomly distributed throughout the ranked gene list or are found more often in the top or the bottom of the list⁷⁹. The complete list of enriched GO terms is presented in Supplementary Data 18.

- Minor comments:

- o On page 9, line 222, the users used “UCEC” without mentioning what it stands for.

- ### Thank you for raising this, “UCEC” is now replaced with endometrial cancer.

- o In all the figures, the font size of legends and axes is very small. It negatively impacts the readability.

- ### We thank the reviewer for pointing this out, we have increased the font size in all figures.

- o Line 146: there is an extra “fig.”

- ### Thank you for spotting this. This has now been corrected.

- o Line 170: missing citation for “previously identified cancer driver genes”.

- ### Thank you, citations have now been added.

End of Response to the Reviewers

We thank the reviewers for the constructive comments which help us substantially improve our manuscript. In addition to the changes described above, we noticed and fixed minor errors from our originally submitted version. We also carefully re-read the manuscript and made some further minor changes to improve clarity and readability. We think the manuscript is now ready for publication, yet please do not hesitate to contact us should any further questions arise or if you think further corrections are necessary.

Sincerely,

João Pedro de Magalhães on behalf of all authors

References

- 1 Brennan, C. W. *et al.* The somatic genomic landscape of glioblastoma. *Cell* **155**, 462-477, doi:10.1016/j.cell.2013.09.034 (2013).
- 2 Verhaak, R. G. *et al.* Integrated genomic analysis identifies clinically relevant subtypes of glioblastoma characterized by abnormalities in PDGFRA, IDH1, EGFR, and NF1. *Cancer Cell* **17**, 98-110, doi:10.1016/j.ccr.2009.12.020 (2010).
- 3 Yan, H. *et al.* IDH1 and IDH2 mutations in gliomas. *N Engl J Med* **360**, 765-773, doi:10.1056/NEJMoa0808710 (2009).
- 4 Cancer Genome Atlas Research, N. *et al.* Integrated genomic characterization of endometrial carcinoma. *Nature* **497**, 67-73, doi:10.1038/nature12113 (2013).
- 5 Ashley, C. W. *et al.* Analysis of mutational signatures in primary and metastatic endometrial cancer reveals distinct patterns of DNA repair defects and shifts during tumor progression. *Gynecol Oncol* **152**, 11-19, doi:10.1016/j.ygyno.2018.10.032 (2019).
- 6 Cancer Genome Atlas Research, N. Comprehensive molecular characterization of gastric adenocarcinoma. *Nature* **513**, 202-209, doi:10.1038/nature13480 (2014).
- 7 Liu, Y. *et al.* Comparative Molecular Analysis of Gastrointestinal Adenocarcinomas. *Cancer Cell* **33**, 721-735 e728, doi:10.1016/j.ccell.2018.03.010 (2018).
- 8 Ciriello, G. *et al.* Comprehensive Molecular Portraits of Invasive Lobular Breast Cancer. *Cell* **163**, 506-519, doi:10.1016/j.cell.2015.09.033 (2015).
- 9 Berger, A. C. *et al.* A Comprehensive Pan-Cancer Molecular Study of Gynecologic and Breast Cancers. *Cancer Cell* **33**, 690-705 e699, doi:10.1016/j.ccell.2018.03.014 (2018).
- 10 Huang, K. L. *et al.* Pathogenic Germline Variants in 10,389 Adult Cancers. *Cell* **173**, 355-370 e314, doi:10.1016/j.cell.2018.03.039 (2018).
- 11 Alexandrov, L. B. *et al.* The repertoire of mutational signatures in human cancer. *Nature* **578**, 94-101, doi:10.1038/s41586-020-1943-3 (2020).
- 12 Alexandrov, L. B. *et al.* Clock-like mutational processes in human somatic cells. *Nat Genet* **47**, 1402-1407, doi:10.1038/ng.3441 (2015).
- 13 Li, C. H., Haider, S. & Boutros, P. C. Age Influences on the Molecular Presentation of Tumours. *bioRxiv*, doi:<https://doi.org/10.1101/2020.07.07.192237> (2020).
- 14 Korber, V. *et al.* Evolutionary Trajectories of IDH(WT) Glioblastomas Reveal a Common Path of Early Tumorigenesis Instigated Years ahead of Initial Diagnosis. *Cancer Cell* **35**, 692-704 e612, doi:10.1016/j.ccell.2019.02.007 (2019).
- 15 Brosseau, J. P. *et al.* Tumor microenvironment-associated modifications of alternative splicing. *RNA* **20**, 189-201, doi:10.1261/rna.042168.113 (2014).

- 16 Yang, L. X. *et al.* Mitogen-activated protein kinase kinase 4 deficiency in intrahepatic cholangiocarcinoma leads to invasive growth and epithelial-mesenchymal transition. *Hepatology* **62**, 1804-1816, doi:10.1002/hep.28149 (2015).
- 17 Modrek, A. S. *et al.* Low-Grade Astrocytoma Mutations in IDH1, P53, and ATRX Cooperate to Block Differentiation of Human Neural Stem Cells via Repression of SOX2. *Cell Rep* **21**, 1267-1280, doi:10.1016/j.celrep.2017.10.009 (2017).
- 18 Danussi, C. *et al.* Atrx inactivation drives disease-defining phenotypes in glioma cells of origin through global epigenomic remodeling. *Nat Commun* **9**, 1057, doi:10.1038/s41467-018-03476-6 (2018).
- 19 Barthel, F. P. *et al.* Systematic analysis of telomere length and somatic alterations in 31 cancer types. *Nat Genet* **49**, 349-357, doi:10.1038/ng.3781 (2017).
- 20 Milholland, B., Auton, A., Suh, Y. & Vijg, J. Age-related somatic mutations in the cancer genome. *Oncotarget* **6**, 24627-24635, doi:10.18632/oncotarget.5685 (2015).
- 21 Chatsirisupachai, K., Palmer, D., Ferreira, S. & de Magalhaes, J. P. A human tissue-specific transcriptomic analysis reveals a complex relationship between aging, cancer, and cellular senescence. *Aging Cell* **18**, e13041, doi:10.1111/accel.13041 (2019).
- 22 Aramillo Irizar, P. *et al.* Transcriptomic alterations during ageing reflect the shift from cancer to degenerative diseases in the elderly. *Nat Commun* **9**, 327, doi:10.1038/s41467-017-02395-2 (2018).
- 23 Zane, L., Sharma, V. & Misteli, T. Common features of chromatin in aging and cancer: cause or coincidence? *Trends Cell Biol* **24**, 686-694, doi:10.1016/j.tcb.2014.07.001 (2014).
- 24 Klutstein, M., Nejman, D., Greenfield, R. & Cedar, H. DNA Methylation in Cancer and Aging. *Cancer Res* **76**, 3446-3450, doi:10.1158/0008-5472.CAN-15-3278 (2016).
- 25 Perez, R. F., Tejedor, J. R., Bayon, G. F., Fernandez, A. F. & Fraga, M. F. Distinct chromatin signatures of DNA hypomethylation in aging and cancer. *Aging Cell* **17**, e12744, doi:10.1111/accel.12744 (2018).
- 26 Dmitrijeva, M., Ossowski, S., Serrano, L. & Schaefer, M. H. Tissue-specific DNA methylation loss during ageing and carcinogenesis is linked to chromosome structure, replication timing and cell division rates. *Nucleic Acids Res* **46**, 7022-7039, doi:10.1093/nar/gky498 (2018).
- 27 Lee-Six, H. *et al.* The landscape of somatic mutation in normal colorectal epithelial cells. *Nature* **574**, 532-537, doi:10.1038/s41586-019-1672-7 (2019).
- 28 Martincorena, I. *et al.* Somatic mutant clones colonize the human esophagus with age. *Science* **362**, 911-917, doi:10.1126/science.aau3879 (2018).
- 29 Martincorena, I. *et al.* Tumor evolution. High burden and pervasive positive selection of somatic mutations in normal human skin. *Science* **348**, 880-886, doi:10.1126/science.aaa6806 (2015).

- 30 Vijg, J. & Dong, X. Pathogenic Mechanisms of Somatic Mutation and Genome Mosaicism in Aging. *Cell* **182**, 12-23, doi:10.1016/j.cell.2020.06.024 (2020).
- 31 Blokzijl, F. *et al.* Tissue-specific mutation accumulation in human adult stem cells during life. *Nature* **538**, 260-264, doi:10.1038/nature19768 (2016).
- 32 Subramanian, A. *et al.* Gene set enrichment analysis: a knowledge-based approach for interpreting genome-wide expression profiles. *Proc Natl Acad Sci U S A* **102**, 15545-15550, doi:10.1073/pnas.0506580102 (2005).

REVIEWERS' COMMENTS

Reviewer #1 (Remarks to the Author):

The authors have done a thorough job addressing the points raised by all three reviewers. This manuscript sheds new light on how age influences the molecular presentation of cancers, sometimes in a surprising manner, and the data will serve as a resource for others in the years to come.

Signed - James DeGregori

Reviewer #2 (Remarks to the Author):

All my questions have been addressed. -Lei Wei

Reviewer #3 (Remarks to the Author):

The authors addressed all my questions.

Authors' response to the Reviewers' comments

Manuscript: NCOMMS-20-34538A

Title: An Integrative Analysis of the Age-Associated Multi-Omic Landscape across Cancers

Dear Editor,

Thank you for considering our manuscript for publication in *Nature Communications*.
What follows are our responses (marked with ###) to the comments from the reviewers.

Reviewer #1 (Remarks to the Author):

The authors have done a thorough job addressing the points raised by all three reviewers. This manuscript sheds new light on how age influences the molecular presentation of cancers, sometimes in a surprising manner, and the data will serve as a resource for others in the years to come.

Signed - James DeGregori

We would like to thank the reviewer once again for your kind words and positive comments.

Reviewer #2 (Remarks to the Author):

All my questions have been addressed. -Lei Wei

We would like to thank the reviewer once again for your constructive questions and suggestions.

Reviewer #3 (Remarks to the Author):

The authors addressed all my questions.

We would like to thank the reviewer once again for your thoughtful questions and suggestions.

End of Response to the Reviewers

We have addressed editorial requests from our previously submitted version. We also carefully re-read the manuscript and made some further minor changes to improve clarity and readability. We think the manuscript is now ready for publication, yet please do not hesitate to contact us should any further questions arise or if you think further corrections are necessary.

Sincerely,

João Pedro de Magalhães on behalf of all authors